# Water vapor stable isotope memory effects of common tubing materials

Alexandra L. Meyer[1], Lisa R. Welp[1]

[1]Department of Earth, Atmospheric, and Planetary Sciences, Purdue University, West Lafayette, 47907, United States

*Correspondence to: almeyer269@gmail.com or lwelp@purdue.edu*

**Abstract.** Water molecules in vapor can exchange with gaseous water molecules sticking to surfaces of sampling tubing, and exchange rates are unique for each water isotopologue and tubing material. Therefore, water molecules on tubing walls take some time to reach isotopic equilibrium with a new vapor isotopic signal. This creates a memory effect observed as attenuation time for signal propagation in continuous stable water vapor isotope measurement systems. Tubing memory effects in $\delta D$ and $\delta^{18}O$ measurements can limit the ability to observe fast changes, and because $\delta D$ and $\delta^{18}O$ memory are not identical, this introduces transient deuterium excess (D-excess, defined as $\delta D - 8* \delta^{18}O$) artifacts in time-varying observations. A comprehensive performance comparison of commonly used tubing material water exchange properties in laser-based measurement systems has not been published to our knowledge.

We compared how a large isotopic step change propagated through five commonly used tubing materials for water isotopic studies, PFA, FEP, PTFE, HDPE, and copper, at two different temperatures and an air flow rate of 0.635 L min$^{-1}$ through approximately 100 feet (~30.5 m) of ¼ in. (6.35 mm) outer diameter (OD) tubing. All commonly used tubing materials performed similarly to each other in terms of attenuation times, reaching $\delta^{18}O$ location adjusted $\delta D$ and $\delta^{18}O$ 95% completion in less than 45 seconds, with slight variations based on temperature. PFA does appear to perform slightly better than the other materials, though memory metric differences are small. A tubing material commonly used in the early 2000's but reported to have memory effects on $\delta D$, Dekabon, was also tested at ambient temperature and changing humidities. Dekabon isotopic equilibrium was not reached until nearly an hour after source transition, much later than $H_2O$ mixing ratios equilibrated. Bev-A-Line XX (used in some soil $O_2$ and $CO_2$ gas studies) was also tested at ambient temperature, but it did not approach isotopic equilibrium until after nearly six hours of testing. Therefore, we cannot recommend the use of Bev-A-Line XX or Dekabon in water vapor isotope applications. Source transition from heavy to light or light to heavy affected isotopic transition speed only in experiments where $H_2O$ ppm$_v$ was changing. While shorter tubing length and smaller inner diameters shortens the delay of signal propagation through the tubing, they didn't greatly change the attenuation curves under these conditions for the current commonly used tubing materials tested. However, in Dekabon, attenuation curves were greatly extended with increased tubing length. Our results show that the commonly used plastic tubing materials tested were not inferior to copper in terms of isotopic memory under these conditions, and they are easier to work with and are less expensive than copper.

## 1 Introduction

In situ laser absorption spectroscopy of water vapor isotopologues has risen in use over the last two decades enabling continuous measurements (Griffith et al., 2006; Kerstel et al., 2006; Lee et al., 2005; Webster and Heymsfield, 2003). All experimental setups inherently attenuate signal variability due to mixing in the analyzer optical cavities and molecular water interactions with surfaces inside the inlet and analyzer system, especially when different $H_2O_v$ concentrations lead to wetting and drying of the tubing walls. A wide range of tubing materials, air flow rates, temperatures, and pressures have been used in experimental setups, which may result in different timescales for signal attenuation (Aemisegger et al., 2012; Galewsky et al., 2016; Griffis et al., 2010; Schmidt et al., 2010; Sturm and Knohl, 2010; Tremoy et al., 2011). As condensation in tubing is a concern due to phase change

isotopic fractionation, many installations heat the tubing above ambient temperature, use a critical orifice at the tubing inlet to drop pressure in the lines, or do both to keep the tubing air temperature above the dew point temperature (e.g. Griffis et al. 2010; Luo et al. 2019).

Initially Synflex 1300 (also known as Dekabon or Dekoron), commonly used in the carbon dioxide and water eddy covariance flux community, was used in water vapor isotope observations (Gupta et al., 2009; Lee et al., 2005; Tremoy et al., 2011). Dekabon is an aluminum tape with an ethylene copolymer adhesive film coated on both sides, rolled into a tube, and bonded with a high-density polyethylene jacket (Goodrich Sales, Inc, 2005; New Line Hose and Fittings, personal communication, April 29, 2024). It was eventually found to greatly attenuate the water isotopic signals (Griffis et al., 2010; Schmidt et al., 2010; Sturm and Knohl, 2010; Tremoy et al., 2011) and is no longer commonly used in water vapor isotope studies.(Griffis et al., 2010; Schmidt et al., 2010; Steen-Larsen et al., 2014; Sturm and Knohl, 2010; Tremoy et al., 2011) Commonly used tubing material types now include copper (Steen-Larsen et al., 2014) and several types of plastic including polytetrafluoroethylene (PTFE, commonly referred to as Teflon) (Griffis et al., 2010; Sturm and Knohl, 2010), perfluoroalkoxy (PFA) (Schmidt et al., 2010; Tremoy et al., 2011), fluorinated ethylene propylene (FEP) (Luo et al., 2019), and high-density polyethylene (HDPE) (Griffis et al., 2010). Some performance testing was conducted, but the details of the experiments and results are sparse (Griffis et al., 2010; Schmidt et al., 2010; Steen-Larsen et al., 2014; Sturm and Knohl, 2010; Tremoy et al., 2011). Fluorinated polymers (FEP, PFA, and PTFE) are commonly used as transfer lines in chemical, pharmaceutical, food processing, and oil and gas industries because of their chemical- and weather-resistance, as well as their non-stick and dielectric properties (Chemours, 2018). These materials have found favor in water vapor isotope applications for the same reasons.

Air tubing choices are important because materials may have different affinities, or degree of attraction, for the isotopologues of water. This affinity causes a delay in the speed at which the isotopologue signals move through the tubing due to exchange rates with water molecules stuck to the walls, called the memory effect. The memory effect is stronger for $\delta D$ compared to $\delta^{18}O$, presumably due to the stronger hydrogen bonding of the molecules containing deuterium slowing tubing wall exchanges (Griffis et al., 2010; Schmidt et al., 2010; Sturm and Knohl, 2010). This can result in false deuterium-excess (D-excess, defined as $\delta D - 8* \delta^{18}O$) anomalies and is important to minimize when D-excess signals are interpreted in quickly changing atmospheric signals (Galewsky et al., 2016; Managave et al., 2016; Salmon et al., 2019; Sodemann et al., 2017). Some studies have suggested that memory effects may be lessened at higher temperatures and faster air flow rates (Griffis et al., 2010; Pagonis et al., 2017).

It is important to minimize isotopic wall effects in the intake tubing lines and other in-line elements positioned before the analyzer to minimize signal attenuation. Five studies previously reporting memory effects of tubing types tested a maximum of three materials at a time and are summarized in Table 1 (Griffis et al., 2010; Schmidt et al., 2010; Steen-Larsen et al., 2014; Sturm and Knohl, 2010; Tremoy et al., 2011). Most concluded that Dekabon was not suitable for water isotope applications but varied in which tubing was preferred across applications. The National Ecological Observatory Network (NEON) selected FEP for their monitoring installations which has not been widely used in reported studies (Luo et al., 2019). In this study, we tested five of the commonly used and reported best tubing types under nearly identical conditions at two different temperatures to determine which tubing

type and temperature combination results in the smallest isotopic signal attenuation. For contrast, we also tested a tubing material known to have memory issues, Dekabon, and Bev-A-Line XX, a tubing not previously used in published water isotope studies but which is increasingly used in soil $O_2$ and $CO_2$ gas studies (i.e. Brecheisen et al., 2019). Note that Bev-A-Line XX has a patented Hytrel® inner lining and is distinct from Bev-A-Line IV which has been used in a few published water vapor isotope studies (Havranek et al., 2023; Lee et al., 2005; Simonin et al., 2013). Because Dekabon and Bev-A-Line XX have extremely slow isotope response times, they were only tested at ambient temperature and with changing water concentrations to demonstrate the source switching in the experimental setup was working properly.

**Table 1.** Literature findings

| Author, year | Materials tested | Isotopes analyzed/goals | Result |
|---|---|---|---|
| *Schmidt et al. 2010 | Stainless steel, PFA, and Dekabon | $\delta D$ and $\delta^{18}O$, Analyzer calibration | PFA better than SS. Both better than Dekabon. |
| *Sturm and Knohl 2010 | PTFE and Dekabon | $\delta D$ and $\delta^{18}O$, Analyzer characterization | PTFE better than Dekabon |
| Griffis et al. 2010 | "Natural colored" HDPE, Teflon (PTFE), and Dekabon | $\delta D$ and $\delta^{18}O$, $\delta^{18}O$ measurements of evapotranspiration in eddy covariance setups | HDPE equal or slightly better than PTFE. Both much better than Dekabon. |
| Tremoy et al. 2011 | PFA and Dekabon | $\delta D$, $\delta^{18}O$, and D-excess, Analyzer characterization and D-excess measurements | PFA better than Dekabon |
| *Steen-Larsen et al. 2014 | Copper, stainless steel, and PTFE | $\delta D$, $\delta^{18}O$, and D-excess, environmental controls on D-excess measurements | Copper better than SS and PTFE. |

*Indicates experimental details and results of source-switching experiments are included in the peer-reviewed published materials.

## 2 Methods

In this study, we tested PFA, FEP, PTFE, HDPE, and copper at ambient and elevated temperatures using self-regulating heat tape. We switched between two isotopically distinct vapor sources to examine memory effects of each material. We also tested Bev-A-Line XX and Dekabon at ambient temperature and at two humidities.

### 2.1 Analyzer

A Los Gatos Research, Inc. (LGR) Triple Water Vapor Isotope Analyzer (TWVIA) Off-Axis Integrated-Cavity-Output Spectroscopy system (OA-ICOS) was used for testing. An external pump (KNF pump, model N920-2.08) was added to the TWVIA to maximize the turnover rate of air inside the analyzer. The TWVIA itself regulates the outflow to maintain a constant internal pressure, resulting in discontinuous (jumpy) flow rates which averaged 0.635 ± 0.006 L min$^{-1}$ at STP with a cell pressure at ~40 Torr. This resulted in an ~4 s mean residence time of sample air in the analyzer. It is typical to average data over an optimum time interval determined by Allan variance testing to

minimize analyzer noise and maximize measurement precision. In this experiment, the objective was to maximize the analyzer response time in order to resolve potential differences in isotopic signal attenuation during travel through inlet tubing. Applying a running mean to the 1 Hz data would have smoothed the response, masking the signal of interest in this study. Measurement uncertainty was estimated using two second Allan deviation which is the lowest time limit of the Allan deviation code output (Guerrier et al., 2020). The Allan deviation at two seconds for δD and $δ^{18}O$ measured over 18 hours at approximately 9,300 ppm produced by a Los Gatos Research Water Vapor Isotope Standard Source (WVISS) was approximately 1.3 ‰ and 0.6 ‰, respectively, propagating to D-excess precision better than ± 3.3‰. To demonstrate this analyzer performance is consistent with other published studies, the full Allan deviation plot of analyzer variance is presented in Fig. S1.

## 2.2 Experimental setup

### 2.2.1 H₂O matched experiments

The memory effect of the tubing material was tested by switching between two sources of moist air with different isotopic values but nearly identical water vapor mixing ratios (~9,200 ppm, Table S1). A LiCor model LI-610 portable dew point generator (DPG) was used to create a vapor of approximately -187 ‰ δD, -25.5 ‰ $δ^{18}O$, and 17.5 ‰ D-excess, measured by the LGR TWVIA without calibration, from water at 5 °C. The second vapor of approximately -32 ‰ δD, -5.8 ‰ $δ^{18}O$, and 14.0 ‰ D-excess was produced by the WVISS, also measured by the analyzer without calibration. DPG-generated vapor isotopic values for the experiments became isotopically enriched over time as water evaporated from the liquid reservoir. Isotopic δD and $δ^{18}O$ transitions were normalized to a one to zero scale to compare across experiments and adjust for small source water and analyzer drift over time. For this reason, further calibration of the isotopic measurements was not needed. Five replicate switches were completed for each experiment where the vapor sources switched approximately every sixty minutes giving sufficient time to reach a new isotopic equilibrium. We focus on data through the first twenty minutes as equilibrium was already established (with the exception of Dekabon and Bev-A-Line XX).

For each experiment, the WVISS programming and internal valve system controlled the switching between the DPG output connected to the WVISS inlet port and the WVISS (Fig. 1) output to the TWVIA. The WVISS was connected to the analyzer by approximately 100 foot (~30.5 m, lengths listed in Table S1) long sections of $^1/_4$ in. (6.35 mm) outer diameter (OD) test tubing for the main experiments. The Swagelok connection to the analyzer included an extra stainless steel union and ~2.5 in. (~6.4 cm) thick-walled FEP to protect the analyzer bulkhead union threads from wear during the experiment, but this addition is not expected to affect the results significantly. Sensitivity to tubing length and inner volume were investigated using a short (62 in. or 1.57 m) and a long (99 feet $^1/_2$ in. or 29.75 m) piece of thick-walled FEP and long piece of thin-walled FEP. Tubing inner diameters (ID, summarized in Table S1) were $^3/_{16}$ in. (~4.76 mm) with the exception of HDPE and thick-walled FEP, which were $^1/_8$ in. (~3.18 mm) ID. Damaged thin-walled FEP tubing was repaired using three stainless steel Swagelok unions and the Dekabon with one, but this is not expected to affect the results significantly.

Tubing and self-regulating heat tape (EASYHEAT ADKS-0500, 100 foot or ~30.5 m roof and gutter de-icing kit) were wrapped in either flexible foam tape (HDPE, PTFE, thick-walled FEP; AP/Armaflex TAP 18230

insulation tape) or rigid foam pipe insulation (copper, thin-walled FEP, PFA; Tundra brand $^1/_2$ in. or 1.27 cm wall). The thermocouple probe was placed inside the insulation on the side of the tubing opposite of the heat tape, about three inches (~7.6 cm) from the end closest to the analyzer inlet. A datalogger recorded the average temperature over the ~10 h experiments. During heated tubing tests, the tubing was allowed to warm up at least an hour prior to measurements to let the tubing moisture equilibrate to the elevated temperature and minimize the effects of degassing water molecules adhered to the tubing from previous experiments. Differences in the insulation properties of the two materials used and likely differences in thermocouple placement relative to unavoidable internal gradients in temperature resulted in differences in average temperatures for each experiment, ranging from 48.6 to 75.2 °C (Table S1). We note that this heating design is commonly used in field conditions and represents likely inlet conditions. However, the lack of uniform temperature control leads to potential temperature-induced differences that are hard to quantify. All heated experiments (average $60 \pm 8$ °C) are significantly warmer than ambient temperature experiments (average $24 \pm 1$ °C). Dekabon was only tested under ambient conditions and thus was not insulated.

Temperature adjusted tubing residence times were $1.0 \pm 0.1$ s for short thick-walled FEP, $19.7 \pm 1.6$ s for long thick-walled tubing (FEP and HDPE), and $44.5 \pm 3.0$ s for long thin-walled tubing (FEP, PFA, PTFE, and copper). Uncertainties in tubing residence time (a few seconds) based on length (a couple inches) and temperature (due to internal gradients and overall temperature fluctuations) were not considered here. Air flow rates through the tested tubing were controlled by the TWVIA itself, making the tubing flow rate as slow as possible and the analyzer flow rate as fast as possible with this set of equipment. The DPG was operated in a continuous fashion, constantly generating humid air. To maintain these constant conditions, a vent was added before the DPG outlet to the WVISS inlet to provide an overflow when the WVISS was pushing its humid air stream to the TWVIA, otherwise the DPG pump would be pushing against a closed valve. A Dwyer rotameter (model number VFB-65-SSV) was used to monitor outflow from the vent. The vent air flow rate is not critical to the tubing tests because it's simply the overflow. An Omega mass flow meter (MFM, 0–30 L min$^{-1}$ range, model FMA1826A) was used to monitor air flow rates downstream of the TWVIA to verify analyzer conditions remained unchanged during the experiments. A Mesa Labs Bios Definer 220 primary flow calibrator (Mesa Labs, Lakewood, CO, 50–5,000 sccm, accuracy $\pm 1$ % of reading) was used to validate the air flow rate through the TWVIA and test tubing at the inlet of the TWVIA prior to the experiments but was not included during the actual experiments. When the primary flow calibrator was removed, no change in the TWVIA outlet flow was detected on the Omega MFM. Rotameter flow rates were verified at the beginning of the experiments using the primary flow calibrator. The DPG vent flow rate was ~0.9 L min$^{-1}$ when the DPG was sampled by the TWVIA and ~1.5 L min$^{-1}$ when the WVISS was sampled, consistent with the 0.6 L min$^{-1}$ flow rate of the analyzer.

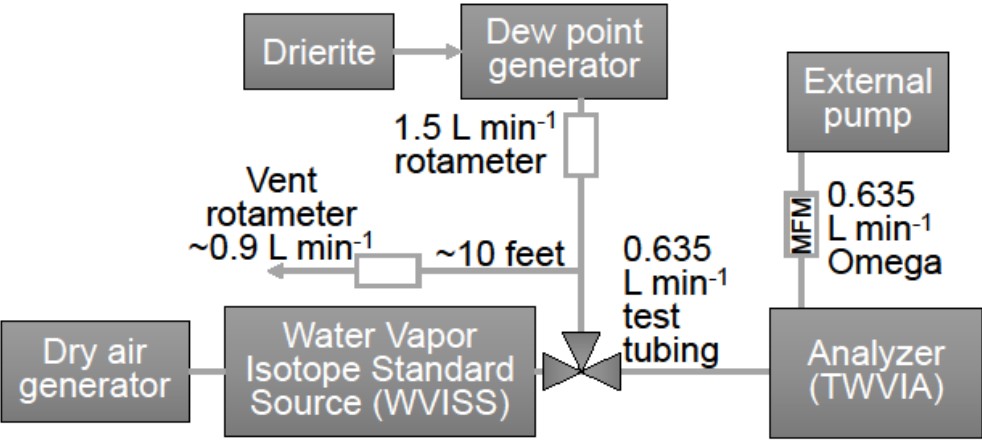

**Figure 1.** Instrument setup for memory effect tests. The WVISS controls switching between WVISS air and dew point generator air (depicted here as an external 3-way valve, but it's internal to the WVISS), which is passed through test tubing of up to 100 ft (~30.5 m) and either heated or unheated to the analyzer. The flow through the test tubing is controlled by the analyzer and the external pump flow rate.

### 2.2.2 H₂O varied experiments

For this set of experiments, the plumbing and flows remained the same. The only difference was this time the two different isotopic sources also had different water vapor mixing ratios (Table S1). The DPG was used to create a vapor of approximately -184 ‰ $\delta D$, -26.2 ‰ $\delta^{18}O$, 25.4 ‰ D-excess, and ~9,300 $ppm_v$ $H_2O$, measured by the LGR TWVIA without calibration, from water at 5 °C. The second vapor of approximately -20.3 ‰ $\delta D$, -8.8 ‰ $\delta^{18}O$, 50.4 ‰ D-excess, and ~16,950 $ppm_v$ $H_2O$ was produced by the WVISS, also measured by the analyzer without calibration. Because data was normalized as above, calibration was not necessary to determine attenuation times. Two to four replicate switches were completed for Dekabon and HDPE tubing depending on the time to reach the new isotopic equilibrium. One replicate of Bev-A-Line XX was run in each direction of the isotopic switch, and results are presented in Fig. S2. Replicate five minute switches comparing the performance of Bev-A-Line XX and HDPE can also be found in Fig. S3.

For each of the H₂O varied experiments, source switching was controlled manually as the TWVIA control of the WVISS unit malfunctioned. The WVISS was connected to the analyzer by approximately 100 ft (~30.5 m, lengths listed in Table S1) long sections of ¼ in. (6.35 mm) outer diameter (OD) HDPE, Bev-A-Line XX, or Dekabon tubing. Other tests were done with a short (~78.7 in. or 2 m) section of HDPE or Dekabon to quantify sensitivity to tubing length and inner volume using high memory materials. Tubing inner diameters (ID, summarized in Table S1) were 0.17 in. (~4.32 mm) with the exception of HDPE, which was ⅛ in. (~3.18 mm) ID. These experiments were conducted under ambient conditions (average 24 ± 1 °C). Temperature adjusted tubing residence times were 2.8 s for short Dekabon, 42.2 s for long Bev-A-Line XX and Dekabon, and 1.5 s and 22.8 s for short and long HDPE, respectively. All other experimental aspects remain the same as detailed in Sect. 2.2.1.

## 2.3 Data processing

Isotopic values were measured at 1 Hz. No calibration to assign values to the international scale was performed on the isotopic measurements because the transitions were normalized to their starting and ending equilibrium values, resulting in signal transitions from one to zero. Isotopic measurements made with this analyzer are known to vary with water mixing ratio and potentially drift over long periods of time. Normalizing the measurements between sources as described below removed any potential influence of instrument or source drift over periods of more than twenty minutes.

For $\delta D$ and $\delta^{18}O$, the individual transitions were normalized from 1 to 0 and then replicates were averaged to characterize the transition memory and uncertainty. Initial $\delta$ values (normalized to one) were either the maximum $\delta$ value after the source switch indicator in the data file (short thick FEP and long Bev-A-Line XX) or the average of five seconds on either side of that maximum $\delta$ value (all other experiments). Final $\delta$ values (normalized to zero) were the average of measurements 600–1200 seconds after the source switch in $H_2O$ matched experiments. Diverse experimental lengths were used during the $H_2O$ varied experiments, with lengths ranging from 336–31,001 s depending on time to equilibrium (Table S1). The variation in experimental lengths resulted in final $\delta$ values ("0") set as the average of at least the last 50 seconds depending on time to equilibrium and length of the experiment (see Table S1 for exact intervals used to average). D-excess was calculated as $\delta D - 8* \delta^{18}O$. D-excess was not normalized in the same way as $\delta D$ and $\delta^{18}O$ because the shape of the attenuation curve is different. First, a 10 s running mean was applied. We then calculated the average D-excess value for each replicate over 600–1200 seconds after the source switch in $H_2O$ matched experiments, and the timespan the final $\delta$ values were averaged over (the last 50 or 100 s) in the $H_2O$ varied experiments. This average was then subtracted from all data points within a replicate to adjust for small changes in D-excess source waters between replicates, especially in the DPG vapor which undergoes evaporative enrichment and D-excess decrease. These timespans after the source switch (600–1200 s for $H_2O$ matched and the last 50 or 100 s for $H_2O$ varied experiments) visually appear to be conditions of tubing equilibration and were used to calculate source vapor sample averages given in Table S1 and summarized in Sect. 2.2. Replicates were screened based on successful WVISS-to-DPG and DPG-to-WVISS switching and consistent water vapor mixing ratios ensuring that vapor source generators were operating properly. Four replicates were discarded from the collected data due to water mixing ratio variability from the WVISS. These discards include one replicate each from heated PFA and 100 ft Dekabon $H_2O$ matched experiment, and two from the 100' HDPE $H_2O$ varied experiment. Last, replicates were averaged to reduce noise.

When comparing experiments between different tubing lengths and IDs, differences in the internal volume result in different tubing residence times due to advection. The flow in all experiments was estimated to be laminar with Reynold's numbers calculated between 579 and 870. In Sect. 3.1 we will describe how the experiments are advection delay-adjusted to compare transitions directly.

Memory analysis included both directions of the isotopic switch. Isotopically enriched-to-depleted (WVISS-to-DPG) figures are presented in the main body of the text, and isotopically depleted-to-enriched (DPG-to-WVISS) transitions are available in the supplemental information (Figs. S4, S5, S6, and S7).

## 2.4 Memory quantification

Memory effects can in some respects be analogous to a low-pass filter, smoothing high frequency variability (e.g. Zannoni et al., 2022). Previous studies have approximated the smoothing of a fast step change input as an exponential transition and report a threshold time to some percentage of completion like an e-folding (63 %), 90 %, or 95 % (Aemisegger et al., 2012; Schmidt et al., 2010; Steen-Larsen et al., 2014; Sturm and Knohl, 2010). In some cases, the threshold metrics were obtained from the data directly (Steen-Larsen et al., 2014; Sturm and Knohl, 2010) and in others it appears an exponential function was fit to the data first and the metrics were extracted from the fit (Aemisegger et al., 2012; Schmidt et al., 2010). A second method used in the literature takes the first derivative of the normalized transition (Steen-Larsen et al., 2014) and characterizes an impulse response function using curve fitting (Jones et al., 2017; Kahle et al., 2018). We have quantified memory effect metrics using both methods.

### 2.4.1 Threshold metrics

We extracted attenuation threshold metrics directly from the normalized and replicate-averaged data (not an exponential fit). An e-folding time corresponds to $\tau = 1/e$ of the signal transition remaining to reach a new value. In this study, we have chosen to estimate attenuation threshold times at $1\tau$ (~63 %) and $3\tau$ (~95 %) completion of the switch to the next $\delta D$ and $\delta^{18}O$ value, denoted as $t_{63\%}$ or $t_{95\%}$ respectively (Schmidt et al., 2010). These $t$ values are the time the averaged curve intersects the threshold percent value. We chose not to fit exponential curves to extract an e-folding time, because the measured attenuation curves were not accurately described by an exponential curve (not shown). The 1 standard deviation envelope was calculated by taking the standard deviation of the two to five replicates at each time step. Errors associated with attenuation threshold times were determined by finding the time that the 1 standard deviation envelope of the averaged replicates intersects the completion threshold. Because the analyzer measures in discrete 1 s intervals, the raw $t_{63\%}$ and $t_{95\%}$ value outputs the next second from where the averaged curve intersects the threshold percent value. This leads to slight differences in $\delta^{18}O$ location adjusted $t_{63\%}$ and $t_{95\%}$ values (discussed in Sect. 2.4.2) compared to the sweepout curves presented in Sect. 3.

D-excess signals of the source transitions are not unidirectional and memory must be quantified differently. Previous studies reported that $\delta D$ signals take longer to equilibrate with the surface of tubing materials compared to $\delta^{18}O$ signals due to isotopic effects of hydrogen binding with the tubing walls (Aemisegger et al., 2012; Griffis et al., 2010; Schmidt et al., 2010; Sturm and Knohl, 2010). The D-substituted hydrogen-bonds exchange with the vapor more slowly. This difference between isotope signal speed leads to a D-excess transition that has a transient anomaly until the $\delta D$ signal propagation catches up to the $\delta^{18}O$ signal. The direction of the D-excess transient peak depends on the direction of the isotopic signal switch. In the enriched-to-depleted transition, the enriched $\delta D$ signal is retained on the tubing walls creating a transient, positive anomaly in D-excess while approaching equilibrium. However, in a depleted-to-enriched transition, the depleted $\delta D$ signal has been preserved on the tubing walls creating a negative D-excess anomaly during isotopic equilibration. The absolute value of the maximum transient peak was identified and associated errors are given as the standard deviation of the replicate D-excess values at the time of the maximum peak (Table S2). The threshold chosen to measure completion in D-excess transitions is a 3 ‰

threshold within the new equilibrium value ($t_{3‰}$). The 3 ‰ threshold is a conservative estimate of analyzer precision of D-excess measurements if δD precision was 1.0 ‰ and δ[18]O precision was 0.25 ‰.

To compare the attenuation threshold times across experiments, we adjusted for differences in signal propagation due to the time it takes air to move through the tubing from the WVISS and mixing inside the analyzer, controlled by the air flow rate through the instrument, optical cavity size, test tubing volume, and air flow rate (Schmidt et al., 2010), as well as temperature. Smaller tubing IDs, increased temperature, and shorter tubing lengths tested here will all shorten lag times associated with a measurement. δ[18]O lag times were calculated via breakpoint analysis to determine the point where slope changes. We created a linear model using the first 300 s of data in most cases after the source switched then utilized the "segmented" function in R's "segmented" package on the time series to find the breakpoint (Muggeo, 2022). In some cases, different observation intervals were used for short FEP and $H_2O$ varied tests, see Table S1 for exact intervals. The breakpoint lag estimates likely have an error of a few seconds. The exact uncertainty was not quantified. Average measured lag times for 100 ft (~30.5 m) thin-walled tubing were 53 s, and 1.5 s for the short thick-walled tubing in the $H_2O$ matched experiments. In the results, the time axis in the plots and quantitative threshold metrics ($t_{63\%}$, $t_{95\%}$ and $t_{3‰}$) in the tables were adjusted by fitted δ[18]O location time (discussed in Sect. 2.4.2) to more easily compare tubing dimension influence on transition smoothing.

**2.4.2 Impulse response method**

In the impulse response method, we take advantage of the first derivative of the observed attenuation curves to clearly identify the timing and rates of change. To decrease the noise in the first derivative, it's necessary to reduce noise in the observed attenuation curves. In previous studies, noise reduction is achieved by fitting a smooth transfer function to the observations. Jones et al. (2017) and Kahle et al. (2018), used a lognormal times lognormal (log-log) function to fit the data, while in Steen-Larsen et al. (2014) only one lognormal is used. For our attenuation curves, neither a single or double lognormal fit the observed data well. Our data was most accurately recreated by a transfer function of the form in Eq. (1) (with the exception of the depleted-to-enriched transition for $H_2O$ matched HDPE, $H_2O$ varied depleted-to-enriched 2 m HDPE, and enriched-to-depleted Dekabon in both sets of experiments where an additional normal fit was added):

$$\delta_{transfer}(t) = c_1 * \left[1 + erf\left(\frac{log(t) - \mu_1}{\sigma_1 \sqrt{2}}\right)\right] * \left[1 + erf\left(\frac{log(t) - \mu_2}{\sigma_2 \sqrt{2}}\right)\right] * \left[1 + erf\left(\frac{t - \mu_3}{\sigma_3 \sqrt{2}}\right)\right] + c_2 \tag{1}$$

where $t$ is time since switching, $\sigma$ is the location of each log/normal, $\mu$ is the standard deviation of each log/normal, and $c_1$ and $c_2$ are scaling factors. The values of $\sigma_1$, $\sigma_2$, $\sigma_3$, $\mu_1$, $\mu_2$, and $\mu_3$ are optimized by minimizing the squares of errors using the "DEoptim" global optimization function in the R package of the same name (Ardia et al., 2022). The form of the fitting model here is not that important as long as the observations are faithfully reproduced in the smooth curve fit, as seen in Fig. 2a.

Once a transfer function is fit, the first derivative of the transfer function is calculated to obtain the impulse function. We fit the impulse function by the model in Eq. (2) based on a skew-normal function added to a normal gaussian function.

$$\delta_{impulse}(t) = \left( c_1 * \left[\left(\frac{1}{\sqrt{2\pi}}\right) * e^{\frac{-x_1^2}{2}}\right] * \left[\frac{1}{2} + erf\left(\frac{x_1 * \alpha}{\sqrt{2}}\right)\right]\right) + \left(\left[\left(\frac{1}{\sqrt{2\pi}}\right) * e^{\frac{-x_2^2}{2}}\right] * c_2\right) \qquad (2.1)$$

$$x_1 = \frac{(t-\xi)}{\omega} \qquad (2.2)$$

$$x_2 = \frac{(t-\mu)}{\sigma_m} \qquad (2.3)$$

where in the skew-normal terms, $\xi$ is the location of the maximum impulse peak, $\alpha$ is shape, and $\omega$ is scale, $t$ is time since switching, $\sigma_m$ is the standard deviation of the additional PDF and $\mu$ is its mean, and $c_1$ and $c_2$ are scaling factors. The parameters are solved for using a two-step method: first using the "DEoptim" function (Ardia et al., 2022) to provide an approximate initial guess, and second utilizing the "nls" non-linear least squares function in the "stats" R package of base R (R Core Team, 2023) to provide parameter fine-tuning and uncertainty estimates of each parameter.

While Jones et al. (2017) was able to fit impulse functions of their data solely with a skew-normal PDF fit (a standard normal probability distribution function times a standard normal cumulative distribution function, or PDF * CDF), we most accurately reproduced the first derivative by adding an extra PDF in Eq. (2). Figure 2b shows a comparison of the Jones et al. (2017) impulse function skew-normal fit compared to the impulse function fit we used in this study. Our impulse function model fits the memory tail in our experiments better than the skew-normal PDF model from Jones et al. (2017).

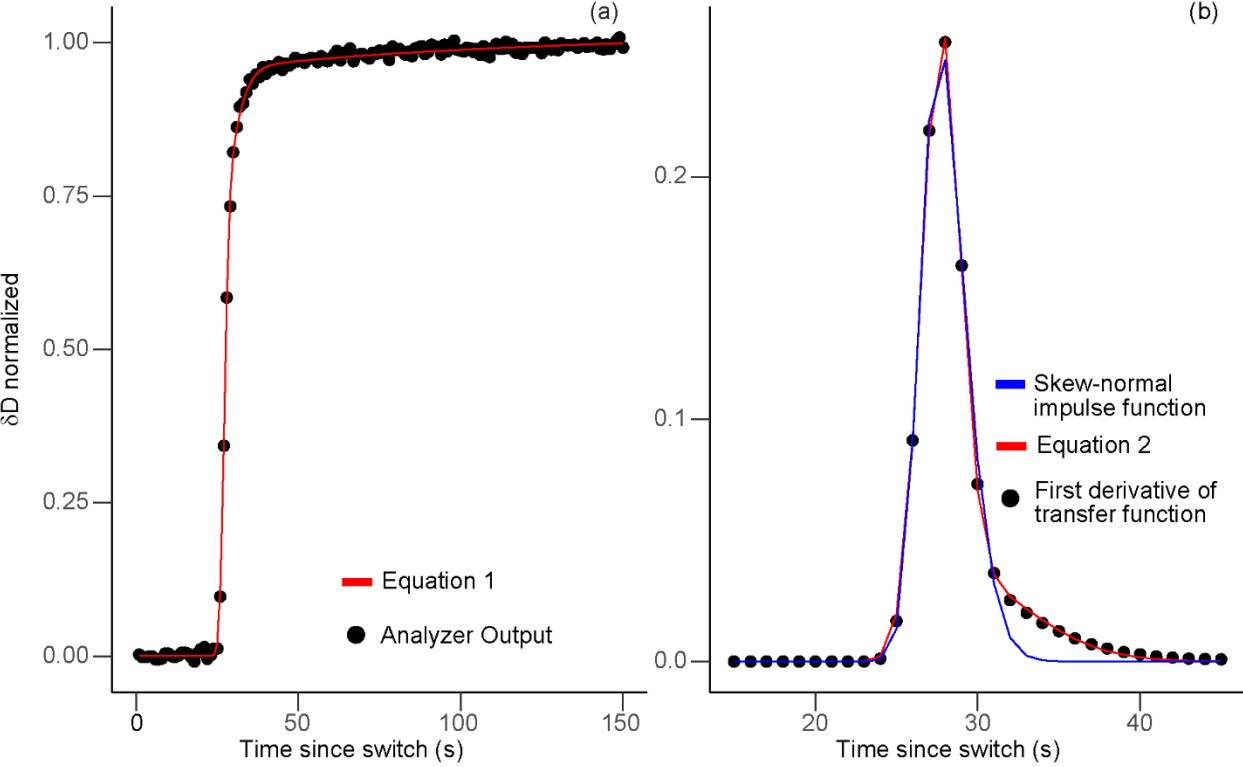

**Figure 2.** Example of model function fits for the unheated long thick-walled FEP experiment. Panel (a) compares normalized and averaged analyzer output (black dots) with the transfer function given in Eq. (1) (red line). Panel (b)

compares the impulse function derived from the first derivative of the transfer function fit evaluated every second (black dots), with the fit from Eq. (2) (red line) and the skew-normal impulse function (blue line) used in Jones et al. (2017) and Kahle et al. (2018).

We extracted two memory metrics from the impulse fitting. First, the skew-normal parameters of shape ($\alpha$, a descriptor of the shape of the curve or other asymmetry of the distribution) and scale ($\omega$, a measure of the spread of the distribution) were used to estimate a mixing time ($\sigma_s$) from Eq. (3). The $\sigma_s$ metric has also been called mixing length in Jones et al. (2017) or diffusive length in Kahle et al. (2018) where analysis time relates distance in the ice cores. The $\sigma_s$ is a metric of how much mixing occurs due to diffusive flow within the tubing. Error for $\sigma_s$ is propagated from the errors associated with shape and scale. Second, we also estimate the standard deviation of the additional PDF ($\sigma_m$) in Eq. (2) critical for fitting the memory tail in the observations which gives additional information about memory not captured by the skew-normal curve.

$$\beta = \frac{\alpha}{\sqrt{1+\alpha^2}} \tag{3.1}$$

$$\sigma_s{}^2 = \omega^2 * (1 - \frac{2\beta^2}{\pi}) \tag{3.2}$$

$$\sigma_s = \sqrt{\sigma_s{}^2} \tag{3.3}$$

## 3 Results

### 3.1 Comparison of residence, lag, and location times in $H_2O$ matched experiments

The residence time of air in the inlets is mathematically predicted using the tubing ID, length, temperature, pressure within the tubing, and air flow rate through the tubing (Table S2). Residence times are decreased by decreasing the

tubing length and inner diameter or increasing temperature through tubing and analyzer, as tested here. Average $\delta^{18}O$ lag times from breakpoint analysis correlate well with predicted residence times (Fig. S8a). For the long thick-walled tubing, the calculated residence time is approximately $19.7 \pm 1.6$ s, with slight variations due to temperature and small length differences which agrees well with observed $\delta^{18}O$ lag of $23.1 \pm 1.2$ s. For long thin-walled tubing, the calculated residence time is approximately $44.5 \pm 3.0$ s, and average $\delta^{18}O$ lag time is $53.0 \pm 4.0$ s (not including

Dekabon, due to instrument malfunction). The largest discrepancies between residence and $\delta^{18}O$ lag times ($< 12.5$ s, with the exception of Dekabon) are found in unheated copper and unheated PFA. Tubing roughness was not considered when calculating residence times, as flow was assumed to be laminar and flow rate was measured at the end of the tested tubing closest to the analyzer and therefore should be representative of the actual flow rate in the tubing. For short thick-walled FEP, the residence time is $1.0 \pm 0.1$ s and average $\delta^{18}O$ lag time is $1.5 \pm 1.7$ s. Overall,

heated tubing lag and residence times were shorter than their unheated counterparts (Table S2).

Similarly, the location time parameter fitted using the impulse response method is the timing of the maximum peak of the impulse function (or the steepest portion of the attenuation curve, discussed in Sect. 2.4.2). The location time is sensitive to the advection lag and the steepness of the isotopic transition. Our estimated $\delta^{18}O$ location time

for the long thick-walled tubing (25.6 ± 1.3 s, Table S2, excluding Dekabon) matches the $\delta^{18}O$ lag time above when accounting for the < 5 seconds between the initial signal change and the maximum slope of the attenuation curve (or peak in the impulse function). Because of this relationship, $\delta^{18}O$ location times correlated well with the observed $\delta^{18}O$ lag times (Fig. S8b), $t_{63\%}$ estimates from the experiments (Fig. S8c, excluding Dekabon), and residence times (Fig. S8d). The differences in location time between different tubing experiments is not fully explained by differences in residence time predictions. The location time extracted from the $\delta D$ impulse function is slightly longer than the location time extracted from the $\delta^{18}O$ impulse function, but they correlate well (Fig S8e). Dekabon $\delta D$ location time is comparatively much longer (~30–50 s longer) than the $\delta^{18}O$ location time (Table S2). Overall, location is closely related to other methods of timing isotopic transitions, including lag, residence time, and e-folding time in the tubing materials we tested, with the exception of Dekabon.

To visualize differences in curve shape between tubing materials tested using different internal volumes (due to length and ID) and air density (due to temperature), a common transition point was defined. This is similar to adjusting by lag time (e.g. Steen-Larsen et al., 2014) or predicted residence times. Given uncertainties in the breakpoint analysis of lag time and tubing temperature uncertainties which influence residence time, we decided the $\delta^{18}O$ location time was the most self-consistent way to collapse the experiments on top of each other in the figures. Adjusting by $\delta^{18}O$ location time also allows comparison to the $H_2O$ varied and Dekabon experiments, as the valve switching time was not precisely recorded by the software.

**3.2 Direction of isotopic and water vapor concentration transitions**

$H_2O$ varied and $H_2O$ matched experiments for 100 ft (30.48 m) HDPE and Dekabon tubing were used to determine if there was a difference in the enriched-to-depleted and depleted-to-enriched switches due to the isotopologues or net degassing of the tubing walls (Fig. 3). $H_2O$ matched Dekabon experiments did not exhibit clear differences depending on the isotopic switch direction, while there were clear differences in $\delta D$ and D-excess depending on switch direction for $H_2O$ varied Dekabon (Fig. 3 panels a, c, and e). $H_2O$ varied Dekabon clearly shows longer $\delta^{18}O$ location adjusted $t_{95\%}$ times in the enriched-to-depleted switch direction than the depleted-to-enriched direction. $H_2O$ matched HDPE also does not exhibit clear differences in switch direction (Fig. 3 panels b, d, and f). $H_2O$ varied HDPE shows a clear difference in $\delta D$ $t_{95\%}$ between switch direction outside of $t_{95\%}$ error, but no clear difference in $\delta^{18}O$ or D-excess. Overall, differences in HDPE threshold metric values are much smaller than the differences between Dekabon values. Impulse response metric patterns for both tubing types are mixed.

There is a difference in the $H_2O$ varied switch direction and little to no clear and consistent difference in switch direction in $H_2O$ matched experiments. Most results presented in the following sections are $H_2O$ matched. We discuss both switch directions in the text, present figures of the enriched-to-depleted switch transition only and place the depleted-to-enriched transition figures in the Supplemental.

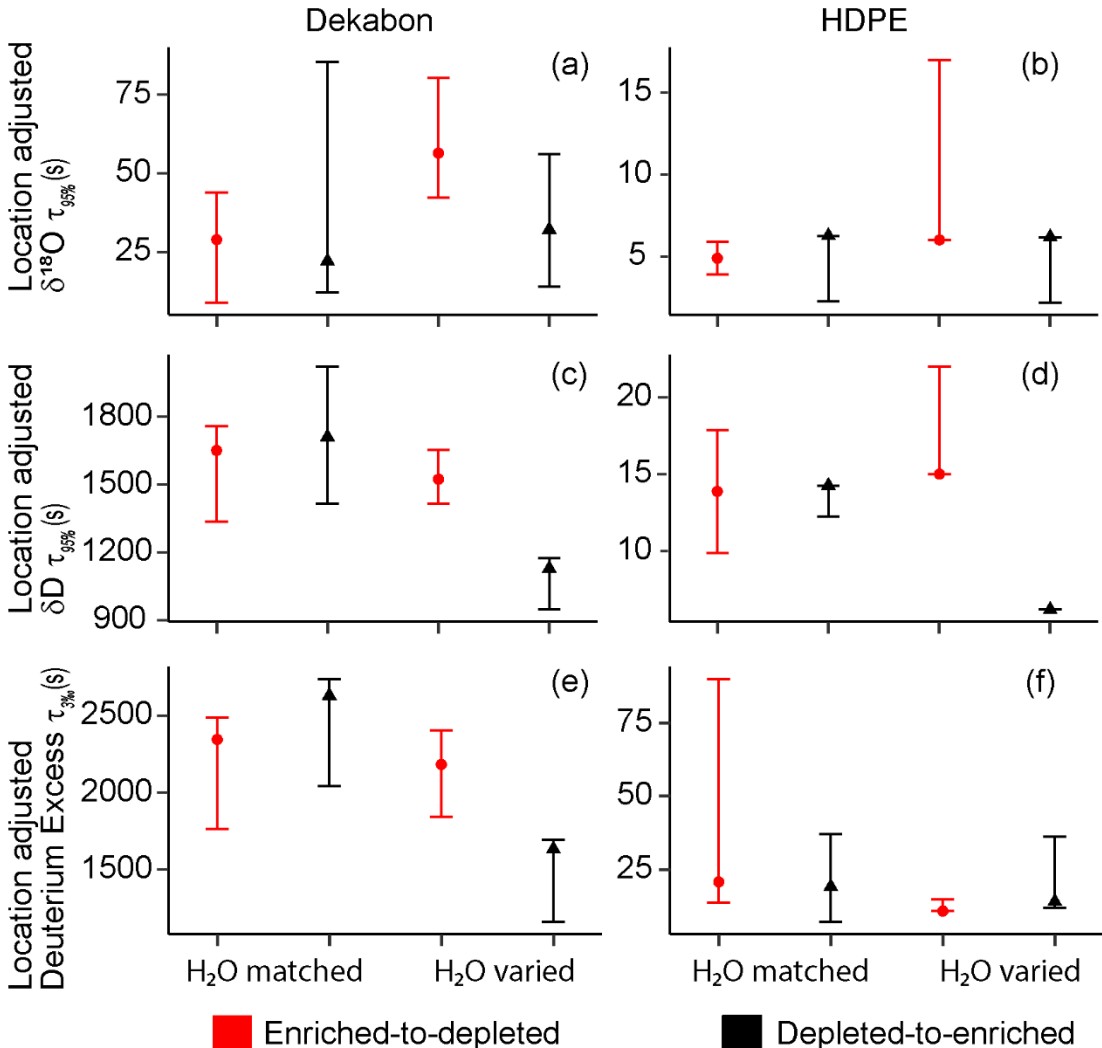

**Figure 3.** $\delta^{18}O$ location adjusted $t_{95\%}$ for $\delta^{18}O$ and $\delta D$ transitions and $t_{3\text{‰}}$ times for D-excess in $H_2O$ matched and varied experiments of isotopic transition directions for long HDPE and Dekabon. The left panels (a, c, and e) depict Dekabon tests, and the right panels (b, d, and f) depict the HDPE tests. The depleted-to-enriched direction is indicated in black. The enriched-to-depleted direction is in red. Error bars indicate the maximum and minimum threshold times from the standard deviation of the experiment replicates. We saw differences in switch direction only in $H_2O$ varied experiments.

### 3.3 Tubing material and temperature

#### 3.3.1 Visual inspection of mean attenuation curves

The mean attenuation curves for the enriched-to-depleted transitions for all $H_2O$ matched long thin-walled tubing experiments and HDPE, which is thick-walled, are compared in Fig. 4 and the depleted-to-enriched results are in Fig. S4. Attenuation curves for each experiment have been adjusted by the $\delta^{18}O$ location time metric to remove the

influence of different air lag times caused by different tubing IDs and temperature sensitive air density to more easily compare equilibration times of different tubing dimensions. Therefore, 0 s in these figures indicates the time of most rapid change in the transfer function and the peak of the impulse function for each experiment. The $\delta$D signal was also $\delta^{18}$O location adjusted to highlight potential differences in equilibration speeds between the two isotopologues. Dekabon stands out as the tubing material with the longest isotopic memory in $\delta$D and $\delta^{18}$O and the largest difference between $\delta^{18}$O memory and $\delta$D memory (Figs. 4 and S4). There are slight variations within the rest of the tubing material type and temperature performances. Specifically, thin-walled FEP $\delta$D results show the next slowest transitions compared to other tubing experiments and indicate FEP has the second largest difference between $\delta^{18}$O memory and $\delta$D memory. The $\delta^{18}$O location adjusted attenuation curves for $\delta^{18}$O have no consistent difference in where they intercept the $t_{63\%}$ threshold between heated and unheated experiments (Figs. 4 and S4).

$\delta$D attenuation times were longer compared to $\delta^{18}$O, creating a transient positive D-excess anomaly in the enriched-to-depleted transitions before equilibrating with the new vapor source isotopic values (Fig. 4). In the enriched-to-depleted transition, propagation of the depleted $\delta$D signal was delayed relative to the depleted $\delta^{18}$O signal (as shown by the orange lines denoting isotopic means of the non-Dekabon tubing in Fig. 4, panels b and d). The $\delta$D signal transition was also delayed relative to $\delta^{18}$O in the depleted-to-enriched transitions (Fig. S4, panels b and d) leading to a transient negative D-excess anomaly (Fig. S4, panels e and f). In Fig. S4, D-excess plots were flipped for easier graphical comparison with the enriched-to-depleted transition. D-excess attenuation times are typically longer than the $t_{95\%}$ times for $\delta$D or $\delta^{18}$O (Table S2) while the slower isotopic propagation of the $\delta$D signal catches up to $\delta^{18}$O. Different D-excess values between experiments in Fig. 4 panels e and f are caused by D-excess drift of the DPG over the experiments.

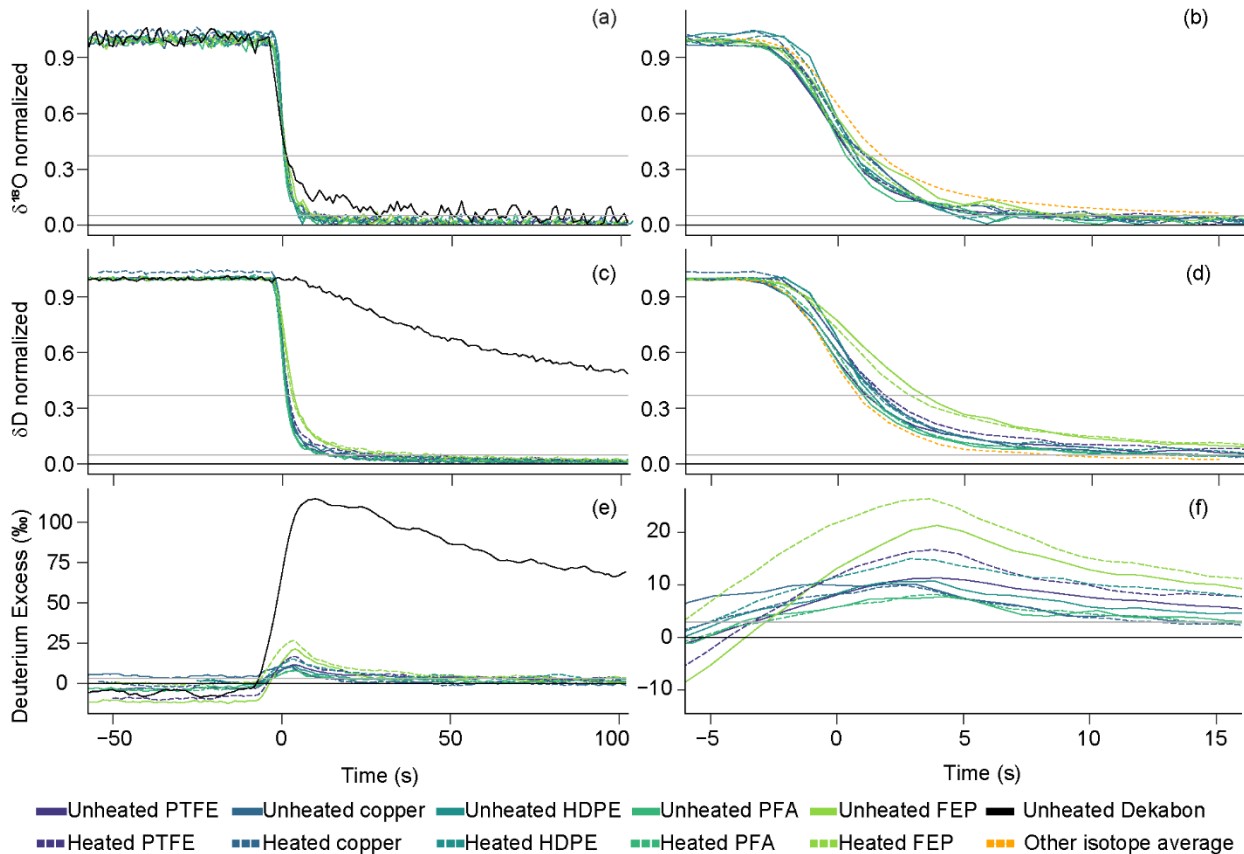

**Figure 4.** Mean attenuation curves for enriched-to-depleted (WVISS-to-DPG) $H_2O$ matched transitions of each tubing type for $\delta^{18}O$ (a, b), $\delta D$ (c, d), and D-excess (e, f) plotted as seconds since the $\delta^{18}O$ impulse function peak (i.e. $\delta^{18}O$ location adjusted time). The first column (panels a, c, and e) depicts time from 50 s before the peak of the $\delta^{18}O$ impulse function for each experiment to 100 s after, while the second column (panels b, d, and f) zooms in on time from -5 to 15 s and excludes the Dekabon results. Solid lines indicate unheated experiments, while dashed lines indicate heated experiments. For $\delta D$, $\delta^{18}O$, and D-excess, only Dekabon and FEP show clear differences in material type, and only FEP shows clear differences in heated and unheated experiments. The full Dekabon attenuation curve can be found in Fig. S9. An orange curve in panel (b) shows mean $\delta D$ of all experiments for comparison with $\delta^{18}O$ and the orange curve in panel (d) shows mean $\delta^{18}O$ for comparison with $\delta D$. These means exclude Dekabon. Here, FEP is long thin-walled as a comparison to the rest of the thin-walled tubings. HDPE is thick-walled and has a smaller ID than the other tubing shown here. Gray horizontal lines indicate thresholds of 95 % and 63 % transition completion for $\delta D$ and $\delta^{18}O$, and 3 ‰ for D-excess, while a black line at zero indicates full equilibrium completion. Depleted-to-enriched results are presented in the supplemental materials.

### 3.3.2 Quantitative memory metrics

After $\delta^{18}O$ location adjustment, there are few consistent differences between heated and unheated tubings when comparing the same material (Figs. 5, S5, and Table S2). While in $\delta^{18}O$ location adjusted $t_{95\%}$ most heated tubing times are similar to or longer than their unheated counterparts, in $\sigma_m$ and $\delta^{18}O$ location adjusted $t_{63\%}$, heated tubing

times are generally similar to or shorter than the unheated. In $\sigma_s$, most signal is within error and there are no consistent patterns between switching direction (Table S2). Overall, heated memory metrics are generally either similar to or smaller than those of the unheated memory metrics when comparing the same tubing types without $\delta^{18}O$ location adjustment (Table S2), with the exception of HDPE (both directions) and depleted-to-enriched PTFE $t_{95\%}$ time.

Each memory metric provides a different ranking of tubing materials based on slight numerical differences in metric values, and all tubings appear operationally similar with the exception of Dekabon (Table S2). Some common patterns in these rankings do emerge in the $H_2O$ matched experiments. Of the tubings we calculated memory metrics for, Dekabon is the worst. The rest of the tubing $t_{95\%}$ and $t_{3‰}$ times are given in Fig 5. We see clusters of tubings that have relatively shorter and longer times to equilibration, specifically in the $\delta^{18}O$ location adjusted $\delta D$ signal, illustrated in Fig. 5b. Thin-walled FEP, HDPE, and PTFE appear slightly slower to equilibrate than the rest of the tubing materials in the enriched-to-depleted direction, while PFA and copper equilibrate slightly faster. Please note that this figure also includes differences in length and inner diameter. While thick-walled FEP is presented here as a direct comparison to HDPE (which is also thick-walled with a smaller ID), the rest of the materials had similar IDs to thin-walled FEP. Comparison of different dimensions of FEP experiments are discussed in Sect. 3.4.

There are differences in relative rankings based on temperature, switch direction, and tubing material type, but these relative rankings vary depending on the memory metric used. Based on $t_{95\%}$, $t_{63\%}$, and $t_{3‰}$ times in the enriched-to-depleted direction, PFA and copper appear similar to each other and slightly better than the rest of the tubing material types. However, according to most impulse response metrics ($\sigma_s$ and $\sigma_m$), HDPE (thick-walled) has the shortest attenuation impulse response time. PFA and copper have the longest $\delta^{18}O$ impulse response times in the enriched-to-depleted direction after Dekabon. For D-excess, the best tubing materials in the enriched-to-depleted direction were copper ($t_{3‰}$) and PFA (by the absolute value of the maximum D-excess peak), while long thin-walled FEP was the worst for resolving D-excess signal (after Dekabon). We did not calculate impulse response metrics for Bev-A-Line XX due to its inferior performance and greatly extended curve shape, but it is clearly an inferior tubing with long memory times (Fig. S2).

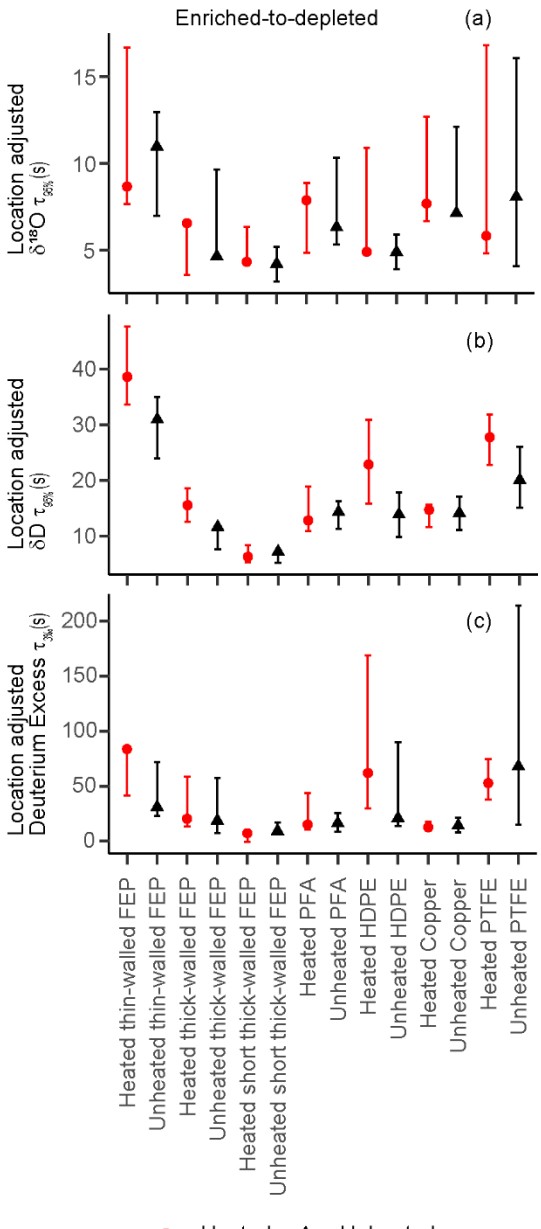

**Figure 5.** $\delta^{18}O$ location adjusted $t_{95\%}$ (panels a and b) and $t_{3‰}$ (panel c) times comparing heated (red) and unheated (black) experiments for all tubing types. The enriched-to-depleted switch direction is depicted here while the depleted-to-enriched transition data is located in Fig. S5. We did not see clear differences in tubing temperature influence, and only very small differences between tubing material type. Thick-walled FEP is presented here as a direct comparison to thick-walled HDPE, while the rest of the materials were thin-walled with larger IDs. Thin-walled, thick-walled, short, and long FEP experiments are shown in Figure 6 and discussed in Sect. 3.4.

## 3.4 Tubing inner volume and length

Properties like tubing length and inner diameter affect mean the transit time through the tubing and the time it takes the signal change to reach the analyzer, but these properties do not appear to greatly influence the shape of the attenuation curve after $\delta^{18}O$ location adjustment in the FEP $H_2O$ matched experiments (Figs. 6 and S6). In these $H_2O$ matched experiments, the short and long thick-walled tubing $\delta^{18}O$ and $\delta D$ transitions overlap each other (Fig. 6 panels b and d), but the long thin-walled tubing has a slightly shallower $\delta^{18}O$ slope (Fig. 6b) and a bigger delay between the $\delta D$ and $\delta^{18}O$ signal transitions (Fig. 6f). While there is not much separation between curves visually, the quantitative memory metrics varied for $\delta D$ with length and inner diameter with longer memory times for longer and larger volume tubing. Short thick-walled FEP in general has smaller memory metrics than long thick-walled FEP, which in turn generally has smaller memory metrics than its thin-walled counterpart. Longer memory metrics were also observed for $\delta^{18}O$ for both switching directions of $t_{95\%}$ and in enriched-to-depleted $\sigma_s$, although other metric differences did not consistently show this pattern

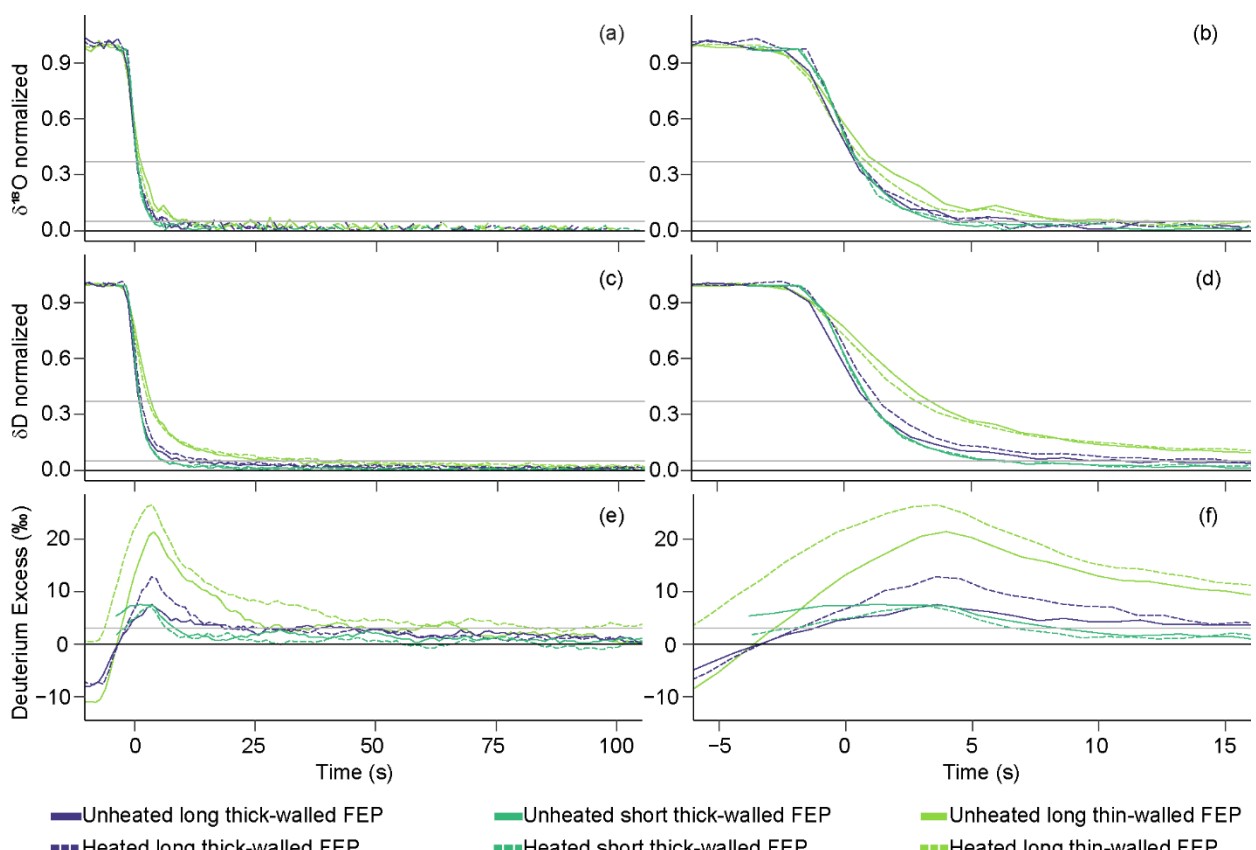

**Figure 6.** Mean attenuation curves for only FEP tubing for enriched-to-depleted (WVISS-to-DPG) transitions comparing tubing length and inner diameter for $\delta^{18}O$ (a, b), $\delta D$ (c, d), and D-excess (e, f) plotted as seconds since the $\delta^{18}O$ impulse function peak (i.e. $\delta^{18}O$ location adjusted time). The first column (panels a, c, and e) depicts time from -5 to 100 s, while the second column (panels b, d, and f) zooms in on time from -5 to 15 s. Solid lines indicate

unheated experiments, while dashed lines indicate heated experiments. Gray horizontal lines indicate thresholds of 95 % and 63 % transition completion for $\delta D$ and $\delta^{18}O$, and 3 ‰ for D-excess, while a black line at zero indicates full equilibration. The $\delta^{18}O$ location adjustment for the short tubing is much shorter than that of the long tubing, leading to a line that appears to start abruptly at approximately -3 s.

In the $H_2O$ varied experiments, tubing length influence on the shape of the attenuation curve after $\delta^{18}O$ location adjustment depends on the tubing material (Figs. 7 and S7). While in HDPE, the short and long tubing isotopic signals are similar to each other in both directions (Figs. 7 and S7, panels b and d), in long Dekabon the isotopic signal transitions are much slower than the short Dekabon in both switch directions. In long Dekabon, the much

shallower $\delta D$ slope (Figs. 7 and S7, panels c and d) and a bigger delay between the $\delta D$ and $\delta^{18}O$ signal transitions leads to a D-excess anomaly of approximately 120 ‰ (Figs. 7 and S7, panels e and f), the largest D-excess anomaly of all tubings tested. This D-excess anomaly is much smaller in short Dekabon (~40 ‰) and demonstrates the more similar signal transitions between $\delta D$ and $\delta^{18}O$. Long Dekabon also has a much shallower $H_2O$ transition slope than the rest of the tubings tested, including short Dekabon, which reacts more similarly to long HDPE when water vapor

concentration is changed (Figs. 7 and S7, panels g and h). Overall, isotopic transitions lag $H_2O$ transitions, as seen when comparing Fig. 7 panels b, d, and h. Short Dekabon consistently has similar or shorter memory metrics than long Dekabon. Short HDPE generally has similar or shorter memory metrics than long HDPE, with the exception of $\delta^{18}O$ enriched-to-depleted $\sigma_m$ and depleted-to-enriched $\delta D$ $\sigma_s$ and $\sigma_m$. Again, we've effectively normalized for tubing length, volume, and temperature through the $\delta^{18}O$ location adjustment, and so differences in the attenuation curve

steepness could be attributed to vapor-wall interactions that are independent of bulk flow.

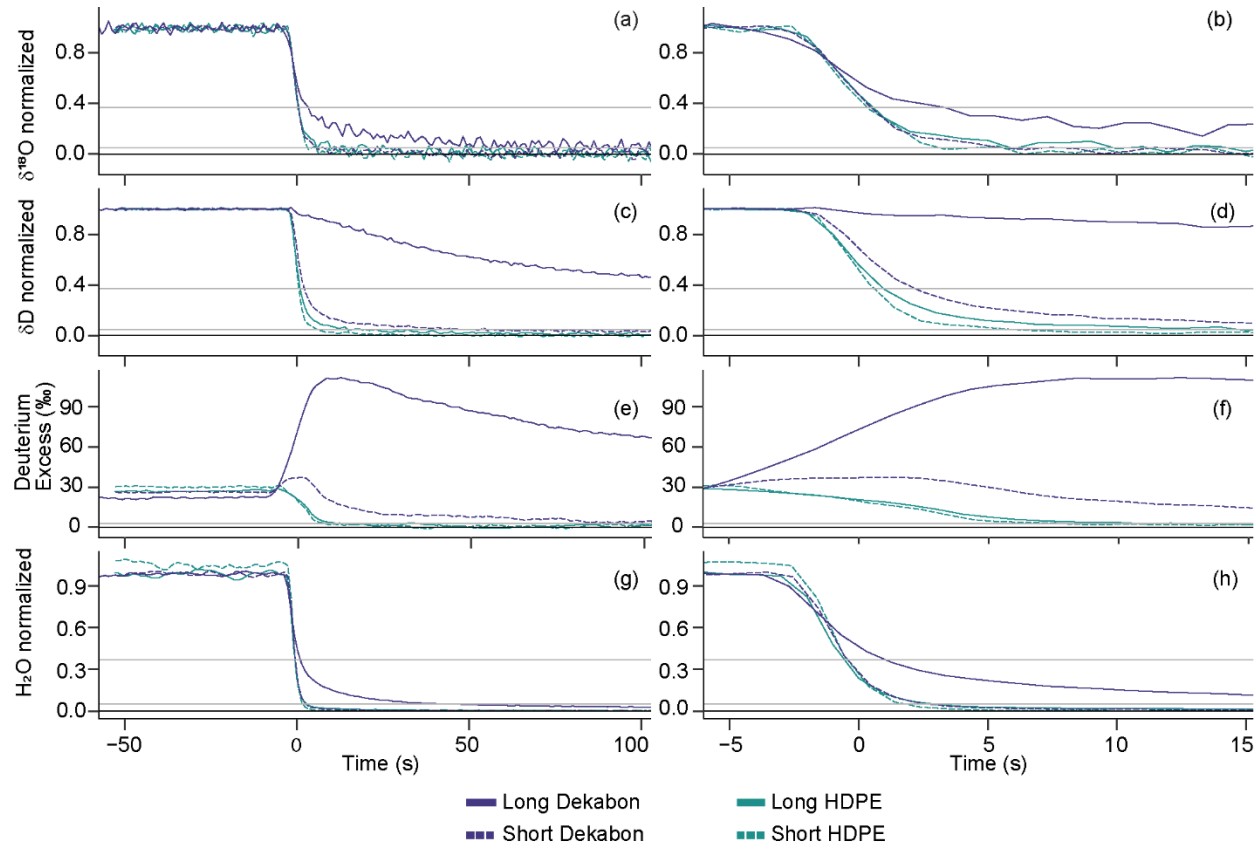

**Figure 7.** Mean attenuation curves comparing length for 2 m (~78.7 in.) and 100 ft (30.48 m) HDPE and Dekabon tubing for enriched-to-depleted (WVISS-to-DPG) transitions for $\delta^{18}O$ (a, b), $\delta D$ (c, d) ), D-excess (e, f), and $H_2O$ (g, h) plotted as seconds since the $\delta^{18}O$ impulse function peak (i.e. $\delta^{18}O$ location adjusted time). The first column (panels a, c, and e) depicts time from -50 to 100 s, while the second column (panels b, d, and f) zooms in on -5 to 15 s. Solid lines indicate 100 ft (30.48 m) lengths, while dashed lines indicate 2 m (~78.7 in.) lengths. Gray horizontal lines indicate thresholds of 95 % and 63 % transition completion for $\delta D$ and $\delta^{18}O$, and 3 ‰ for D-excess, while the black line at zero indicates full equilibration.

## 4 Discussion

Previous water vapor isotope studies have tried to identify suitable tubing material to use in sample inlets, and several different materials have been used. To our knowledge, the results of rigorous testing for wall adsorption/desorption effects leading to memory artifacts have not been published. Theory based on principles of gas chromatography and gas-wall partitioning predicts that the residence time of gases adsorbed on tubing walls is linearly proportional to tubing inner diameter and length and should decrease at higher temperatures as gas saturation concentration changes (Pagonis et al., 2017). The experiments performed in this study begin to test these predictions for water vapor isotopes.

**4.1 Review of material properties**

We hypothesized that predictions of tubing material performance could be made based on tubing material properties. Out of material properties commonly reported by manufacturers, we selected two properties we thought may play a role in fractionating wall effects: water absorption % by tubing weight and relative permittivity. Hydrophobic materials that are nonpolar and have a high relative permittivity (also known as the dielectric constant, or a material's ability to prevent electrical fields from forming) should be ideal for water vapor isotope studies as polar

water molecules are affected by and can induce electric fields (Aemisegger et al., 2012). As previously shown, $\delta D$ signal transitions are slowed compared to $\delta^{18}O$ signals, due to isotope-dependent hydrogen-bonding interactions with tubing walls. Limiting these interactions should lead to reduced isotopic attenuation times. Material specifications vary by manufacturer and material purity, but in general, FEP and PTFE materials are expected to have the least amount of water absorption of the tubing types we tested (Table 2). The inner liners of Dekabon and

Bev-A-Line XX are proprietary information and/or non-disclosable, and as such the information found in Table 2 was not available upon request. However, plastic polymers do have the capability to absorb water at hydrophilic sites and potentially in free volume within the polymer up to ~12 % by material weight for ethylene-vinyl alcohol copolymers (Cava et al., 2006), as discussed in Sect. 4.5. Metals have a relative permittivity value of approximately one due to their sea of electrons, which in this case interact with the polar water molecules. Larger values of relative

permittivity are better in this case, as water vapor molecules will be less attracted to the metal. HDPE, FEP and PTFE have the highest ability to prevent electrical fields. FEP and PTFE may be expected to have the shortest isotopic attenuation times based on combined water absorption percentage and relative permittivity.

At the air flow rate we tested, the isotopic memory metrics of FEP and PTFE were not noticeably superior to the other tubing tested. If the material properties listed here correspond to a fractionating effect, their impact may be

too small to measure, possibly due to the additional ~4 s residence time of the analyzer optical cell and internal plumbing. Alternatively, the material properties listed may impart non-fractionating effects. Slower tubing air flow rates and faster analyzer flow rates may result in more precise resolution of memory differences between tubing types, but further research would be needed to determine if these or other material properties affect water isotope memory.

**Table 2.** Material properties of tubing type options and their water absorption percentages and relative permittivity values.

| Material | Water absorption % by tubing weight | Relative permittivity (Dielectric constant) @ 1 MHz ($\varepsilon_r$) |
|---|---|---|
| FEP | <0.01[a] | 2.1[b] |
| PFA | <0.03[a] | 2.05–2.06[b] |
| PTFE | <0.01[a] | 2.0–2.1[b] |
| HDPE | 0.10[a] | 2.3–2.4[b] |
| Copper | N/A | ~1 |

[a]after being submerged for 24 hours (ASTM D570). This metric is solely for plastic materials. [b](Electrical properties of plastic materials, 2021)

### 4.2 Direction of isotopic and water vapor concentration transition

Quantitative memory metrics were used to determine if there was a difference in the enriched-to-depleted and depleted-to-enriched switches. We focused on the $t_{95\%}$ threshold metric, as the $t_{63\%}$ values and impulse metrics were too small to gain a complete understanding of any differences in either switch direction or between $H_2O$ matched

and varied experiments. Similar to Aemisegger et al. (2012) and their tests with PFA, we found the enriched-to-depleted switch exhibited longer attenuation times during $H_2O$ varied experiments (Fig. 3 and Table S2). However, we were not able to replicate this finding in the $H_2O$ matched experiments. While Aemisegger et al. (2012) indicated that they found a difference in switch direction regardless of the span of isotopic transition, they do not mention testing the effects of varying the span of water vapor concentration. We posit that the difference in isotopic

transition dependent on direction is actually a dependence on the wetting and drying of the tubing and analyzer walls. Related to Aemisegger et al.'s (2012) claim that isotopic adsorption (depleted-to-enriched switch) is faster than the desorption process in the heavy isotopologues, this may be a side effect of the water vapor transition because it is energetically harder to pull a water vapor molecule off a tubing wall and replace it than it is to simply add more molecules. $H_2O$ concentration variation between sources is likely the driving factor of memory metric

differences based on transition direction.

### 4.3 Effects of material and temperature

We found $\delta D$ and $\delta^{18}O$ attenuation curves between the commonly used tubing materials were slightly different, but operationally similar, at the flow rate, humidity, and temperatures tested (Figs. 4, 5, S4, and S5). Dekabon and Bev-A-Line XX attenuation curves were much longer. Our results are consistent with Griffis et al.'s (2010) assertation

that HDPE is similar to PTFE. We were not able to replicate Steen-Larsen et al.'s (2014) finding that copper was better than PTFE in all metrics. In our study, tubing materials performed similarly when comparing all memory metrics: $\sigma_s$, $\sigma_m$, $t_{63\%}$, $t_{95\%}$, $t_{3‰}$, and the absolute value of the maximum D-excess peak. However, given differences in D-excess values between sources, we caution overinterpreting the maximum D-excess anomalies between experiments, as evidenced by the different starting points in Fig. 4e. After accounting for differences in tubing ID

and length, PFA seems to be one of the better tubing materials by a very small margin.

Warmer temperatures are theoretically predicted to reduce attenuation times (Pagonis et al., 2017) by changing the saturation concentration of gases. Additionally, the lower molar density of the warmer air means there is a shorter residence time through the tubing, increased molecular movement, faster wall exchanges, and warmer tubing material means fewer molecules are stuck to the tubing walls. We found some evidence of reduced attenuation times

620 in heated experiments in comparing $\delta^{18}O$ location times and $\sigma_m$ from the impulse function method (Table S2). $\delta^{18}O$ location times for heated tubings are always shorter than their unheated counterparts, and $\sigma_m$ values are similar to or

shorter for heated tubings in most cases. Calculated residence times and observed lag times were also shorter for heated tubings, but to varying degrees depending on the tubing. By $\delta^{18}O$ location adjusting the threshold memory metrics, we are effectively removing the effects of temperature on residence time of air in the tubing. The removal of temperature effects on residence time is why the differences in Figs. 5 and S5 are not consistent between heated and unheated. However, $\delta^{18}O$ location adjusting may also remove some wall effect differences between materials, and limits our ability to discuss $t_{63\%}$ given the similarity with location times discussed in Sect. 3.1. The memory tail, best described by $t_{95\%}$ (Figs. 5 and S5) and $\sigma_m$ (Table S2), shows that heated experiments are not consistently slower or faster to equilibrate than their unheated counterparts. Overall, heating the tubing to avoid condensation does not negatively impact the isotopic measurements. Similarly, Aemisegger et al., (2012) found little difference in attenuation times with varying PFA tubing temperatures.

**4.4 Effects of tubing inner volume and length**

The model in Pagonis et al. (2017) indicates that tubing delays are expected to scale proportionally with tubing length and diameter. The difference in length in the thick-walled FEP long and short experiments was a factor of 19 (99 ft/5.2 ft, or 30.2 m/1.6 m). Both HDPE and Dekabon had a factor of 15 times difference in length between the long and short experiments (100 ft/78.7 in or 30.48 m/2 m). While the results show a clear influence of longer memory times in longer tubing compared to short tubing, we were unable to find quantitative evidence of linear dependence on memory metrics like $t_{95\%}$. Though there are slight differences in the memory metrics we calculated, this is likely due to the influence of the analyzer. Because the analyzer optical cavity and inner tubing has a residence time of ~4 s, we are unable to resolve the residence time ($1.0 \pm 0.09$ s) and memory metrics associated with the short FEP tubing only. Even with the large length difference in FEP, the shape of the isotopic attenuation curves remained similar after location adjustment which removes the length-based residence time differences (Figs. 6, 7, S6, and S7). $\delta^{18}O$ location time adjusted $\delta D$ $t_{95\%}$ and $t_{63\%}$ times for long thick-walled FEP tubing were at maximum 6.2x and 2.6x greater than the short, respectively. The mixing time scales ($\sigma_s$) and the memory tail metric ($\sigma_m$) both showed less than a doubling between short and long tubing. These modest differences in wall-effect memory metrics may be because the analyzer memory itself makes it impossible to accurately isolate and quantify the short tubing response.

From the $\delta^{18}O$ location adjusted comparison of the same material (FEP) with different IDs (Figs. 6 and S6, panels a and c), we conclude that a bigger ID causes the increased memory. In our experiments, ID increased by a factor of 1.5x between thick- and thin-walled FEP ($^1/_8$ in. or ~3.18 mm ID compared to $^3/_{16}$ in. or ~4.76 mm ID). There was clear separation in Fig. 6 between thick- and thin-walled long FEP even after isolating the memory tail by adjusting for bulk delay differences with the $\delta^{18}O$ location adjustment. The thin-walled FEP had a less steep slope and longer $t_{63\%}$ intercept than the thick-walled tubing. $\delta^{18}O$ location time adjusted memory metrics also show a slight increase in memory with ID increase, with an average 2.1x larger memory metric for $\delta D$ and 1.98x larger memory metric for $\delta^{18}O$ between thin- and thick-walled long FEP tubing (Table S2). The long thin-walled FEP consistently showed the slowest $\delta D$ signal transitions of the FEP tubings tested (Figs. 6 and S6, panels c and d), as

well as of all the tubing materials tested (Figs. 4 and S4, panels c and d). We also note that PTFE and PFA tubing had the same ID as the thin-walled FEP In the $H_2O$ matching experiments, $^3/_{16}$ in. (~4.76 mm). PTFE and PFA experiments showed a shorter attenuation threshold time than the thin-walled FEP, and longer attenuation times than tubing with smaller IDs like HDPE (Figs. 5 and S5). Therefore, the results in Figs. 4 and S4 must be evaluated while considering ID differences.

In summary, we found that all tubing dimensions, including ID and length, had some effects on the threshold metrics (Figs. 6, 7, S6, and S7) even after removing differences in $\delta^{18}O$ location times in signal propagation to the analyzer based on tubing inner volume and the temperature influence on molecular density and the total number of molecules in the tubing. While these overall memory metric differences exist, they are small in the materials and dimensions tested, with the exception of Dekabon. The operational impact among commonly used ¼ in. (6.35 mm) OD tubing inlets is expected to be limited. Tubing length and ID play a role in our experiment, consistent with theory that tubing length, ID, and material properties such as density and partitioning depth will affect the attenuation time of chemical compounds on or in a tubing wall (Pagonis et al., 2017). Further tests under faster analyzer and slower tubing air flow rates would be needed to further validate whether these influences are linearly proportional.

## 4.5 Relative attenuation time differences between δD and δ¹⁸O

δD signals have been demonstrated to take longer than $\delta^{18}O$ signals to isotopically equilibrate with tubing materials due to isotope-dependent hydrogen-bonding interactions with the tubing walls (Griffis et al., 2010; Schmidt et al., 2010; Sturm and Knohl, 2010). Our results confirm these previous findings. The speed difference between isotopic signal propagation has been reported as a ratio of attenuation times between the slower δD signal and the faster $\delta^{18}O$ signal. Published results show 1.4–3.5 times greater attenuation time for δD signals than $\delta^{18}O$ signals depending on tubing air flow rate, tubing type, humidity, and memory metric used (Aemisegger et al., 2012; Griffis et al., 2010; Schmidt et al., 2010; Zannoni et al., 2022). As demonstrated in the results, different metrics show different sensitivities to different parts of the attenuation curve. The threshold metrics we calculated are most similar to the quantification metrics used in earlier studies and our results (excluding Dekabon) have similar ranges. For $t_{63\%}$, this ratio ranges from 1.0–3.46, and for $t_{95\%}$ 1.0–4.79. For $\sigma_m$, a metric that we expected to be sensitive to the characteristic long δD memory tail, δD values were 0.9–1.7 times longer than $\delta^{18}O$ values. Overall, δD signals are slower than $\delta^{18}O$ signals in all tubing materials.

Dekabon attenuation metric δD/$\delta^{18}O$ ratios are greater than those of the other tubing materials; signal ratios are up to 14.1 for $\sigma_m$, 61.5 for $t_{63\%}$, and 71 times longer for δD vs $\delta^{18}O$ $t_{95\%}$ values across all 100 ft (30.48 m) experiments. We speculate that the long equilibration time and large memory metric ratios in Dekabon and Bev-A-Line XX are due to a large number of water molecules bound to the tubing surface and an affinity to bind to water molecules containing deuterium. These tubing materials may have larger free volume in their molecular structure, an increased number of hydrophilic sites on the surface, or both (Cava et al., 2006), which allows for more water molecules to be held on or in the tubing. To estimate the amount of water Dekabon might hold, we estimated a mass-balance based tubing reservoir size using the location-adjusted δD $t_{95\%}$ value as a residence time for water

molecules in the tubing. This resulted in a reservoir of 0.13 g $H_2O$, or approximately 0.02% water absorption by tubing weight. This weight percentage is similar to the upper estimate for PTFE, FEP, and PFA (Table 2). While we

were unable to confirm the exact composition of the inner liner of Dekabon, it is an ethylene copolymer, potentially similar to the ethylene-vinyl alcohol copolymers (EVOH) tested in Cava et al., (2006). Applying the Cava et al. (2006) value of 2 % absorption by weight when our experimental humidity (~30 %) is considered, the amount of water Dekabon could hold is approximately 14.6 g, much greater than the estimated reservoir size based on isotopic residence time. For Bev-A-Line XX, the mass balance reservoir estimate is 1.7 g, or 0.2 % water absorption by

tubing weight. This percentage is double the value for HDPE in Table 2. Because we know that polymers of the same family with different molecular structures can have different affinities and hold varying amounts of water as in Cava et al. (2006), it stands to reason that the Dekabon and Bev-A-Line XX tubings have different affinities and hold different amounts of water than the other tubing materials tested.

**4.6 Fitting attenuation curves**

The overall attenuation curves of the tested tubing material types, lengths, and temperature conditions had effectively the same reverse sigmoidal shape after fitted $\delta^{18}O$ location time adjustment, though in Dekabon this shape is elongated. Previous studies approximated the attenuation transfer function as an exponential curve (Aemisegger et al., 2012; Schmidt et al., 2010; Sturm and Knohl, 2010), similar to the exponential decay response that would be expected for the residence time distribution function of a continuously stirred reactor (Toson et al.,

2019). We found the exponential function was not a satisfactory fit to our experimental observations. A more appropriate mixing analogy could be the axially dispersed plug flow (ADPF) model (Huang and Seinfeld, 2019), as this better matches the reverse sigmoid curve we observe. In the ADPF model, there is a bulk flow that has a diffusive "head" that diverges forwards and backwards from the bulk flow, leading to the observed smoothing of the output signal of an input step-change. This diffusive "head" effectively "smears" the observed isotopic signal. While

the shape of this transfer function seems appropriate, the Huang and Seinfeld (2019) model does not consider gas-wall exchange effects. The transfer function model we introduce here fits the observations sufficiently well, but more work is needed to match the formulas with mixing theory.

Likewise, the impulse fitting method we used is more complicated than previously used (Jones et al., 2017; Kahle et al., 2018). We were able to estimate a mixing time metric ($\sigma_s$) from the skew-normal and a memory tail

metric ($\sigma_m$) from our modified impulse function fitting method. We believe these metrics are signals of diffusion mixing and isotopic wall effects. Mathematically describing the influence of isotopic wall effects using a transfer and impulse function is potentially useful for correcting out memory effects in water vapor isotope measurements, as suggested by Massman and Ibrom (2008) and others (e.g. Aemisegger et al., 2012; Steen-Larsen et al., 2014). Similar corrections have been achieved in the ice core and liquid water isotope analysis communities (e.g. Jones et

al., 2017; Kahle et al., 2018; Vallet-Coulomb et al., 2021). We found more complicated transfer and impulse function models were necessary to fully capture the memory effects in the vapor inlet system compared to the mostly liquid inlet systems described before (e.g. Jones et al., 2017; Kahle et al., 2018; Vallet-Coulomb et al., 2021).

This manuscript should provide a starting point for future work removing the low-pass filter effects on continuous water vapor measurements.

## 5 Implications for measurements

Longer attenuation times smooth signal variability and mask high-frequency features, as shown in Sect. 3.4. While lag times aren't inherently bad (as long as measurements lag in tandem), signal smoothing caused by memory effects will draw out the memory tail and muddle environmental signals. Therefore, the magnitude and speed of atmospheric signal variability as well as the analyzer and sample intake smoothing are important considerations when planning for ambient water vapor isotopic measurements. We found very small differences among commonly used tubing materials under the experimental conditions tested here. While different analyzer air flow rates are not presented in this study, it is known that analyzer flow rate strongly influences sample residence time in the optical cavity of these analyzers and the speed of signal transitions. The Aemisegger et al., (2012) findings that attenuation times were controlled more by analyzer residence times than PFA intake tubing is supported by the results presented in this study.

Prior research clearly identified Dekabon tubing as unsuitable (Griffis et al., 2010; Schmidt et al., 2010; Sturm and Knohl, 2010; Tremoy et al., 2011), a conclusion which was validated in this experiment. Bev-A-Line XX also performed particularly poorly, and we cannot recommend the use of either tubing in water vapor isotopic studies. We also suggest testing the effect of any in-line elements like flow meters, mass flow controllers, or filters on isotopic signal attenuation, especially if they are made from materials not tested in this study. Our experience found a mass flow meter that introduced a large memory effect (not presented here).

### 5.1 Low atmospheric variability measurements

For stationary measurements with one intake and high air flow rates, tubing selection among commonly used materials is not as much of a concern as air advecting past the intake typically changes slowly compared to tubing attenuation time scales we quantify here. Conroy et al., (2016) for example, observed vapor on Manus Island, Papua New Guinea that changed by 22.3 ‰ in $\delta^{18}O$ and 154.8 ‰ in $\delta D$, with the largest change being ~25 ‰ $\delta D$ over a duration of a few hours. The instant isotopic step change in our experiment (19.7 ‰ in $\delta^{18}O$ and 155 ‰ in $\delta D$ for $H_2O$ matched experiments) is extreme compared to typical atmospheric variability at a stationary inlet. For stationary measurements, any of the tested tubing materials besides Dekabon and Bev-A-Line XX should be suitable and would not be expected to produce large transient D-excess artifacts due to memory differences between $\delta D$ and $\delta^{18}O$.

### 5.2 High atmospheric variability measurements

For measurements that need high temporal resolution of small atmospheric isotopic variability like flux gradient and eddy covariance setups or airborne observations, extra precautions should be taken. Griffis et al. (2010) used

spectral analysis in their eddy covariance experiments to show that tube memory effects weren't a concern for $\delta^{18}O$ signals at tubing air flow rates of 12 L min$^{-1}$ and analyzer air flow rates of 1.5 L min$^{-1}$. However, one can't extend that conclusion to slower air flow rates and analyzer residence times should be compared across analyzer types.

Aircraft campaigns are a special concern as they observe not only at high temporal (and spatial) resolution but encounter large and rapid isotopic and humidity variability as well. Especially when conducting vertical profiles, 765    isotopic compositions can vary by hundreds of per mil in $\delta D$. Salmon et al. (2019) found $\delta D$ signal values ranging from -400 to -175 ‰ $\delta D$ within an ~5 minute vertical profile descent between 1200 to 400 m above ground. Similarly, Sodemann et al. (2017) reported flight sections with >200 ‰ $\delta D$ variations in under 5 minutes. While data was collected at 1 Hz, their reported data is a 15 s average, which allows them a 975 m horizonal and 75 m vertical resolution (Sodemann et al., 2017). However, that best-case estimate is based on the data averaging interval 770    and does not consider signal attenuation due to tubing isotopic memory or mixing in the optical cavity (Sodemann et al., 2017). Additionally, averaging over long time periods may not remove D-excess memory bias depending on patterns of increasing or decreasing delta values. The wetting and drying of the measurement system during flights with large changes in altitude, and therefore atmospheric specific humidity, may also increase isotopic attenuation times.

In both eddy covariance and aircraft measurement situations, one might consider increasing air flow through the analyzer and intake tubing and shortening the length of tubing from an intake pickoff point to the analyzer in slow analyzer flow setups as has been suggested in previous studies (e.g. Griffis et al., 2010). While high air flow rates can easily be achieved in the air intake main lines in both high-frequency measurement situations, the air flow rate through the analyzer is typically limited by the analyzer design and control software. If tubing or in-line elements 780    like mass flow controllers affect the speed at which the isotopes are transmitted from the intake to the optical cavity, signals are effectively low-pass filtered (Zannoni et al., 2022). Our experiments show shorter memory effects for shorter tubing compared to longer tubing. Therefore, it is also important to minimize the length of tubing from the intake pickoff point to the analyzer to reduce the residence time of air in the low-flow portion of the system. These considerations should also maximize D-excess data resolution.

**5.3 Liquid water measurements**

Liquid water isotope analysis is also plagued by memory effects when samples are converted to the vapor phase for laser-based spectral isotopic analysis, especially in applications measuring samples with large isotopic differences in the same batch. Common protocols recommend multiple replicate injections and discarding the first few to remove carryover from the previous sample (Coplen and Wassenaar, 2015; IAEA, 2009; Penna et al., 2012). In both OA-790    ICOS and cavity ring-down spectroscopy, Penna et al. (2012) found that when measuring samples with large isotopic differences, up to eight out of eighteen injections had to be ignored to limit memory effects. When analyzing highly depleted Antarctic samples ranging from −231.7 ‰ to −421.1 ‰ for $\delta D$, memory effects of up to 14 ‰ were found in the first injection compared to the "true" value. Liquid water analysis is one example of a case where air flow rates and temperatures of transfer lines are often fixed by the instrument design. Material properties 795    inside the analyzer are important, but this study finds little difference between commonly used material types.

Waiting for equilibrium in the optical cavity may minimize the memory effect, but a time-efficient method to increase sample throughput is to mathematically correct for these repeatable effects rather than attempting to minimize them (e.g. de Graaf et al., 2020; Vallet-Coulomb et al., 2021; Hachgenei et al., 2022). Or, in the case of de Graaf et al., (2020), one can measure small vapor samples on a background of humid air to reduce memory effects.

Work is also being done in the ice core community to correct for signal mixing based on transfer function fitting methods (e.g. Jones et al., 2017; Kahle et al., 2018). These memory correction approaches may provide examples of methods to reconstruct input signal variability from smoothed continuous vapor isotope measurements as well.

**6 Conclusions**

We tested the water isotopic exchange properties of PFA, FEP, PTFE, HDPE, copper, Bev-A-Line XX, and Dekabon. The commonly used materials tested here (not including Bev-A-Line XX and Dekabon) perform similarly. It does not seem necessary to standardize materials used to measure stable water vapor isotopologues to make accurate and comparable measurements in most situations, when using analyzers with similar residence times. We cannot recommend Bev-A-Line XX or Dekabon for use in water vapor applications due to extremely long

attenuation times. At this relative humidity of ~33 %, warmer temperatures did shorten the residence time, lag, and location metrics of the impulse function and $t_{63\%}$ threshold times across all long tubing experiments, but results were not always consistent for $t_{95\%}$. While heating the tubing makes the isotopic signal move through the setup faster due to an air density decrease, heating did not decrease isotopic signal smoothing in all cases, and after $\delta^{18}O$ location time adjustment there were no consistent differences with temperature. However, heating to avoid condensation

does not seem to negatively impact the isotopic measurements. While differences may be found among tubing material types at lower or higher humidities, these experiments are beyond the scope of this study. The direction of the isotopic step change in source transitions affected isotopic transition speeds only in experiments where $H_2O$ $ppm_v$ was changing. Larger tubing IDs and lengths were predicted to increase memory metrics proportionally based on gas-wall partitioning theory (Pagonis et al., 2017), and we found that increasing tubing ID and length increased

the threshold metrics after removing differences in $\delta^{18}O$ location times. The effect of tubing length was most noticeable between 100 ft (30.48 m) and 2 m Dekabon. The other tubing experiments here showed overall memory metric differences do exist, but that they are small in the materials and dimensions tested. In experimental settings, operational impact among commonly used ¼ in. (6.35 mm) OD tubing inlets is expected to be limited.

Researchers must understand the limitations of the air flow conditions and wall effects of their instrumental and

825 intake setups to limit signal memory effects, especially if low air flow rates are a constraint or if there are large isotopic variations over short periods of time. Our experience and results from other published studies indicate that maximizing air flow rates through the analyzer is the most effective way to minimize memory effects when accurate high-frequency D-excess measurements are desired. Our results show that these plastic tubing materials are not inferior to copper in terms of isotopic memory under the tested conditions, and they are easier to work with and are

830 less expensive than copper. As with most decisions, environmental conditions, cost, and preference may influence the type of tubing selected.

**Code/Data availability**

All figure data and scripts, as well as an example workup code, are available at https://doi.org/10.4231/Y13T-6775
(Meyer and Welp, 2024).

**Author contributions**

ALM and LRW designed the experiments and conducted them. ALM adapted code from LRW and added to it for this project, as well as analyzed data. ALM wrote the manuscript draft. ALM and LRW edited the document.

**Competing interests**

The authors declare that they have no conflict of interest.

**Acknowledgements**

We thank Meghan Brown for their assistance in conducting experiments. We thank Matthew Binkley (MS Materials Engineering) for valuable discussion of material properties.

**Financial support**

AM was supported by a Purdue Doctoral Fellowship and Graduate School Summer Research Grant, an Indiana Space Grant, and the National Science Foundation Graduate Research Fellowship Program under Grant No. (DGE-1333468). Any opinions, findings, and conclusions or recommendations expressed in this material are those of the authors and do not necessarily reflect the views of the National Science Foundation.

**Review statement**

We thank the editor Thomas Röckmann and three anonymous referees, as well as community commenter Jonathan Keinan, for their time, suggestions for improvement, and patience.

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
