# Peer review of "Water vapor stable isotope memory effects of common tubing materials"

_Atmospheric Measurement Techniques, 2023_

## Author Comment (AC1)

Author's note to editor: We learned a lot about these experiments from the first set and agreed with the reviewers' concerns that the experimental design could be improved. The revised manuscript is based on the following major changes.

1. We repeated the experiments using a new design. The air flow through the analyzer was increased to maximum flow rate of the external pump and we let the air flow through the tubing be controlled by the analyzer/external pump flow rate. This achieved the shortest turnover time of air in the analyzer and the slowest flow rate through the test tubing given our equipment which should allow the best conditions to identify differences between tubing material types and factors like length and inner diameter.
2. We added a curve fitting data analysis approach to help resolve memory differences between experiments. The new impulse function curve fitting analysis provides two additional memory metrics we discuss in the manuscript. We also kept the attenuation threshold time metrics and added another metric of the D-excess transient anomaly- the absolute value of the maximum D-excess signal peak.
3. We added a new tubing material, Bev-A-Line XX, which demonstrates a much longer memory effect and therefore provides an unacceptable material example.

These major changes mean that not all the previous reviewer comments are now applicable and are listed as such. Author responses are included in red text.

We appreciate the time, suggestions for improvement, and patience of editor Thomas Röckmann, the three anonymous referees, as well as Jonathan Keinan.

Reviewer 1 comments

This article compares the memory effect and lag times associated with laser-based water isotpic measurements for different tubing materials and dimensions. This article will be very useful for people involved in laser-based water isotopic measurements. The article is well written. My comments are relatively minor.

Note that I'm not an expert in laser-based isotopic measurements. Although I enjoyed reviewing this article, the comments by referees who are actually hands-in with such measurements will be very useful and probably more relevant than mine.

General: For readers who are not familiar with feets and inches, it would be helpfull to systematically add between brackets the lengths in international units.

International units are added.

Figure 3 caption: d and e are not described in the caption. Try something like "Mean attenuation times t95% for δ18O (a) and δD (b) and t63% for δ18O (d) and δD (e) and t3 ‰ for D-excess (c)"?
Also in this figure, how were the error bars estimated? Explain this somewhere in the data analysis section?

This figure has been removed in the revision. These memory quantification metrics and new ones are in the supplement. Error quantification is included in the methods section 2.4.1 specifically for threshold metrics, and 2.4.2 for impulse function metrics.

Section 4.2: I might be missing some basic elements to understand this section. Maybe giving a few more sentences of background or explanation would be useful:

o   aren't the tubing cylindrical? If so why aren't the surfaces and volumes linear with length?

This discussion has been removed. Surface area and volumes are linear with length, but in that previous discussion, ID was also changing between short and long tubing.

o   "the shape of the isotopuc attenuation curves remained similar": is it just the shape that remains similar? It looks like it's more than the shapes, the attenuation times remain similar as well, and this looks like the most important result.

Correct, and that is our first sentence of that paragraph. L318-319: "Once adjustments were made for the lag times in signal propagation to the analyzer, we found that the tubing dimensions including ID and length had little effect on the isotopic attenuation times (Figure 4 and S3)."

o   l 327: "However…": why does it contradict the hypothesis that the isotopic memory mainly comes from the analyzer cavity?

No longer applies after we reduced the memory of the analyzer in the new experiments to better see differences between tubings.

o   Why do the rates depend on the fraction of water adsorption sites that are out of equilibrium, rather than the number of sites? Could you give a simple equation (e.g. for the

first-order kinetic reaction) that would allow readers not familiar with this literature to understand this paragraph?

This discussion has been removed. There are now resolvable, but small, differences changing length and ID.

I 399: ". ." -> "." Corrected
* * *
Reviewer 2 comments

The manuscript by Meyer & Welp is a study aimed to show how different tubing materials affect the water vapor isotopic signal propagation inside tubings. The authors tested different kind of tubings by forcing the experimental setup with large isotopic step changes without changing the water vapor mixing ratio. The authors then discuss the shape of rising/falling edges and the timing characteristics of the step change curves (lag, rising time, $t_{63}$, $t_{96}$ etc). The results show very similar characteristics for all the tubing materials tested, regardless of temperature (tested at ambient temperature and 60°C). This study can be highly relevant for the water vapor isotope community, since there is no clear evidence/agreement on what type of tubing is best suited for high frequency atmospheric measurements of water vapor isotope composition. In general the paper is well written and enjoyable to read. Results and concepts are clearly presented and discussed. Figures are of good quality and easy to interpret. However, there are some aspect of the design of the study and choices that I believe the authors must explain/address before the paper is accepted for pubblication. In conclusion, the manuscript requires a major revision in my opinion.

Major comment #1: I am not an expert of OA-ICOS but usually such instruments are equipped with large optical cavity. I will assume an optical cavity volume of ~830 ccm, following Aemisegger et al. (2012) . This volume is ~1.5 times larger than the inner volume of the largest tested tubes (100 feet, $^3/_{16}$" ID).  Moreover, the flushing rate of the instrument is $^1/_3$ of the flow rate in the tubings under test (in fast mode). Therefore, the experiment setup allow to spot only differences at very low frequencies. Indeed, all the high frequency components of the step change are dampened because of long average displacement of water molecules in the optical cavity. Therefore, the conclusion that all the tested tubing types are OK for water vapor analysis is valid only for low frequency analysis (e.g. hourly observations) but not for high-frequency

analysis (e.g. flux, aircraft etc). Since I don't know the characteristics of the TWIA I might be wrong. In case the cavity volume is smaller, please do not consider this comment.

We agreed with the reviewer and decided to repeat all the experiments using the shortest possible residence time of the TWVIA (~4s, Sect. 2.2). We hope the reviewer finds the results of this effort worthwhile.

Major comment #2: The plot reported in the supplement (S2) shows an unusual increase of the Allan variance at short averaging time (>~60 seconds). If the water isotope source is stable (invariant isotope composition with time) and the measurement system is stable (measurement noise is mostly white, instrumental drift is small) the minima in the curves should be found at longer averaging time and the drift (the increase in the adev curve) should be smaller. See e.g. Fig.7 in Aemisegger et al. (2012) or Fig.3 in Jones et al. (2017). **This suggests that the target value of the step change is not stable (i.e. the target isotope value is changing with time in the time frame of the analysis ~1 hour).** This might be due to the change in isotope composition in the source of water vapor? The authors already identify the DPG as a potential source of isotope variability. A correction of the source isotope composition using Rayleigh distillation might be necessary (mentioned at L119-120). It is not clear how large the fractionation of the standard water was during the tests .

Allan plot now presented shows data using the WVISS source which does not trend over time like the DPG source due to the nature of the vapor production. The stable 'end points' are now defined as the average of 600-1200 sec (10-20 min) to minimize the influence of any potential drift. This is a much faster stabilization than the previous experiments, where the 'end points' were 3400-3600 sec due to the slower analyzer flow rates. The new Allan plot shared in Figure S4 shows that the variance doesn't increase until ~200 sec in d18O and 1000 sec in dD which is more in line with expectations. Visual analysis of the hour-long experiments suggests that the drift is minimal within a sweepout, even with the DPG.

Because we normalize the transitions, it's not necessary to correct for small changes in the DPG source water over time. All source water values (uncalibrated) are given in Table S1. Source value differences change by 155.43 ± 3.43 ‰ (~2.2% variance) δD, 19.839 ± 0.463 ‰ (~2.3 % variance) δ18O, and 4.04 ± 3.610 ‰ (~89% variance) D-excess between experiments. See revised section 2.3 for a discussion of why calibration, water mixing ratio dependence, and instrument and source drift corrections are not necessary.

Major comment #3: Please consider to change the step change into the impulse response by computing the derivative (see e.g. Jones et al., 2017, Steen-Larsen et al., 2014). This will let you to discuss how the signal is attenuated by e.g. fitting a normal distribution and looking at the standard deviation of the distribution, which is an indication of the average diplacement of molecules inside your measurement system. For water vapor stable isotope analysis usually the impulses are not symmetrical, therefore a best fit of a log-normal distribution or of an exponentially modified Gaussian distribution should to the job.

We took the reviewer's suggestion and used Jones et al., 2017 as inspiration. The impulse responses of the data were not normal/gaussian nor skew-normal. They had longer memory tails than those models could fit. We fit a combination of a skew-normal and gaussian distributions to fit the experimental results and now report these metrics in the revised manuscript.

Minor comments:

- It is not clear how the start of the step change is detected. I think the swithcing of the 3-W valve is logged but how you detect the "start" of rising-falling edge to precise measure the lag?

The lag is the difference between the time the valve switched (recorded) and the time it takes the signal to propagate through the tubing to the analyzer which depends on length, ID (inner volume), and flow rate. We now utilize the location metric from the impulse response method as a metric to line up the sweepout curves. This is not the 'start', but it correlates well with a lag estimate from a breakpoint analysis also described in the revised manuscript.

- A spectral analysis of the impulse response could be beneficial for understanding the limits of each tubing material for each application (e.g. by identifying the 3dB attenuation and the passband)
For our tubing comparisons, adding another memory metric to compare tubing performance like the 3dB attenuation and passband from a Fourier Transform does not immediately appear to add much additional value on top of the metrics we do provide, including the new impulse response function fit also suggested by reviewers. The 3dB attenuation and passband may be very valuable when characterizing a specific inlet + analyzer design in field studies and for developing back correction methods in the future, as suggested by Steen-Larsen et al., in their 2014 paper.

Steen-Larsen, H. C., Sveinbjörnsdottir, A. E., Peters, A. J., Masson-Delmotte, V., Guishard, M. P., Hsiao, G., Jouzel, J., Noone, D., Warren, J. K., & White, J. W. C. (2014). Climatic controls on water vapor deuterium excess in the marine boundary layer of the North Atlantic based on 500 days of in situ, continuous measurements. _Atmospheric Chemistry and Physics, 14(15), 7741–7756. https://doi.org/10.5194/acp-14-7741-2014

- L185 does this means that the impulse response of your system is guassian? Or at least, symmetrical?

Edited. The impulse response is shown in figure 2. It is not gaussian nor symmetrical.

- L344 Fairly slow? In respect to stable isotope analysis?

Correct- previous work has suggested the faster the air flow rate through tubing and analyzer, the less memory effect. Tubing air flow rates are often over 1 L/min and analyzer flow rates are generally 'as fast as possible'. Our new experiments minimize this concern.

- L494 Link to code/data is not working. Fixed

References

Aemisegger, F., Sturm, P., Graf, P., Sodemann, H., Pfahl, S., Knohl, A., & Wernli, H. (2012). Measuring variations of δ 18O and δ 2H in atmospheric water vapour using two commercial laser-based spectrometers: An instrument characterisation study. _Atmospheric Measurement Techniques_, _5_(7), 1491–1511. https://doi.org/10.5194/amt-5-1491-2012

Jones, T. R., White, J. W. C., Steig, E. J., Vaughn, B. H., Morris, V., Gkinis, V., Markle, B. R., & Schoenemann, S. W. (2017). Improved methodologies for continuous-flow analysis of stable water isotopes in ice cores. _Atmospheric Measurement Techniques_, _10_(2), 617–632. https://doi.org/10.5194/amt-10-617-2017

Steen-Larsen, H. C., Sveinbjörnsdottir, A. E., Peters, A. J., Masson-Delmotte, V., Guishard, M. P., Hsiao, G., Jouzel, J., Noone, D., Warren, J. K., & White, J. W. C. (2014). Climatic controls on water vapor deuterium excess in the marine boundary layer of the North Atlantic based on 500 days of in situ, continuous measurements. _Atmospheric Chemistry and Physics_, _14_(15), 7741–7756. https://doi.org/10.5194/acp-14-7741-2014
* * *
Reviewer 3 comments

The manuscript of Meyer and Welp details a comparison experiment of tubing types at two temperatures. The work deals with the common issue of memory effect in water isotope analysis and aims to minimize this effect by finding the most appropriate material. While in general the experimental setup is logical and the text reads well, **I have some major points that I feel are not addressed well. Also, the text and figures need refining to more clearly communicate the findings.**

*Major comments:*

My main concern about this paper is highlighted in figure 4, where fast and slow analyser flow modes are compared. Based on the text, fast analyser mode increases the flow of air through the optical cavity (x2.5). During fast analyser flow, the MFM was removed, but test tubing was kept. Also, flow rates upstream of the analyser before the venting T, passing through the tested tubing itself, remained constant. Given that your study was designed to test tubing attenuation, and nothing changed in the tubing or flow through the tubing, no difference would be expected between slow and fast analyser modes. Still, Figure 4 indicates a 10x smaller memory effect duration for fast analyser flow compared to slow analyser flow. This suggests that all (equally large) attenuation times found in slow analyser mode, the mode in which all tubing was tested, were predominantly caused by attenuation in the instrument or, as the authors suggest, in the MFM. Thus, not by tubing itself. **How can reliable conclusions be drawn on tubing material type then?**

We agree with the reviewer's concerns and repeated the experiments.

I would highly recommend including dekabon as an additional tubing material. In the introduction you clarify that it is known that dekabon causes attenuation. If you can show it also does using your setup, you can be more confident about the attenuation times you find for the other materials. In the current state, open questions about the setup cast doubt on your finding that attenuation times are independent on tubing material and temperature.

We did not have Dekabon on hand and it is not commonly used in the field anymore. We did include a test of Bev-A-Line XX which showed the extremely long memory time the reviewer wanted to see.

Intuitively, tube length and tube ID impact attenuation, as indicated by your measurement setup. You don't observe this and defend your findings claiming the exchange is a "first-order kinetic reaction" (Sect 4.2). I miss a simple, interpretable, explanation on this, given the context

of tube attenuation, possibly including a figure. Please clarify in your explanation why the following train of thought would be incorrect:

- o with 20x longer tubing, 20x more exchange sites are present, all occupied by isotopic composition 1 just before the switch.
- o after the switch to isotopic composition 2, exchange sites are swapped with a constant rate for each isotopologue, independent of tubing length.
- o as the longer tube has a 20x higher net amount of isotopic composition 1, it will take longer for the output signal to consist of 95% isotopic composition 2.

The reviewer makes logical comments on how to think about this. We don't disagree. It's clearly not a sequential process that all the sites at the beginning of the tubing have to exchange before ones at the end of the tubing. The new experiments with the shorted analyzer residence time are now able to resolve small differences due to length and tubing ID. They still seem surprisingly small, and we do not have a good explanation for why.

You indicate in your introduction (e.g. L. 60) that temperature and air flow rate (and tubing material) have a known "great effect" on attenuation based on various previous studies. Still, your results replicate none of these effects (effective air flow rates through tubing material was changed by wall thickness variations in your experiment). I feel that section 4.1 and 4.2 don't currently provide convincing arguments for why you don't find the known dependencies with temperature or flowrate.

This wording has been removed. In the previous experiments, the analyzer memory was sufficiently large to limit the resolution of tubing differences. We repeated the experiments using faster flow through the analyzer and slower flow through the tubing and found slight effects of temperature and effective flow velocities. See revised Section 3.

***Minor comments:***

**Graphics**

Fig 1.    Please add the flowrate coming from the WVISS. Is this exactly 1.1L/min? Otherwise, why doesn't it need an overblow? Also, given that only mass flow meters (not controllers) are used according to the scheme and text, how is the 1.1 L/min set. Explain in the text if the rotameters were used to set the flows.

New experiments removed all in-line elements including rotameters, MFM, and filter, and the flow rate through the tubing and the analyzer is now set by the analyzer/external pump, which is at maximum speed. Flows were verified at the start of the experiments using a Bios Definer 220.

Fig 2.   Add a theoretical e-folding time based on sample cell mixing. I derive a 25sec residence time based on a 500ml cell (which I think your LGR has) with 100ml/min flow. It shows the reader whether analyser mixing can cause attenuation (seems non-dominant) but generates questions on how the analyser flowrate adjustment has such a large effect.

The previous experimental design had a long analyzer residence time compared to the tubing residence times, so we repeated the experiments using improved conditions to address reviewer concerns. We've included a calculation of the analyzer (sample cell) residence time which is ~4s in the new experimental design (based on an estimated 0.925L volume at 40 Torr and 0.635 L/min flow rate). The short thick-walled FEP tubing has a residence time of ~1.2s. Long thick-walled tubing (different materials with 1/8 inch ID) has a residence time of 22.7 ± 0.2 s, and the long thin-walled tubing (different material with 3/16 inch ID) has a residence time of 50.8 ± 0.8 s.

Fig 4.    This figure should be remade. The panel labels are in the wrong order compared to the description. The y-label of current panel b is wrong (delta D). The legend is unclear as it looks like one large legend while each column has its own.

This figure has been replaced with the new experimental results and care has been taken to ensure the labels are correct

Tab S1. Tubing lengths are only occasionally mentioned, include this everywhere and make units uniform (foot or '). Also, I notice that the WVISS dilution setting was not constant, seemingly affecting the H2O concentration of the mixture generated, why wasn't the dilution constant?

Tubing lengths have been added to the supplemental table S1. The nebulizer flows seem to change slightly over time, likely due to partial clogging. If the dilution flow was changed, it was to match the WVISS output H2O mixing ratio between experiments. In this round the nebulizer was cleaned at one point and replaced with a new nebulizer at another time so dilution flow was adjusted at times. The goal was to keep H2O mixing ratios as close to constant as possible across experiments.

**General**

I noticed frequent incoherent sentence structures, sometimes making it challenging to get the point. I recommend going over the document to improve this.

We have attempted to simplify and clarify sentence structure.

You are in a low flow laminar regime (Re << 3000) through a long tube, yet within-tube flow rate differences, which are roughness dependent, were not explicitly considered as a cause for signal attenuation. I think it would be a valuable point to add (even if effects are non-dominant).

We have calculated Reynold's numbers for the tubing diameters and reported these in the paper and edited the discussion to include the potential effect of tubing roughness.

"The flow in all experiments was laminar with Reynold's numbers calculated between 579 and 870." section 2.3

"The overall attenuation curves of the tested tubing material types, lengths, and temperature conditions had effectively the same reverse sigmoidal shape after location adjustment. The slight differences in signal attenuation could be due to errors in normalization and location adjustments between experiments, differences in tubing internal roughness, and analyzer noise." section 4.5

**Text**

L.104    "fast" and "slow" analyser suggests you used multiple, which I understand was not the case. Also, it would help the reader to explicitly state whether the flowrate through the tested tubing changed (I understand it didn't ) under both analyser regimes. Lastly, why multiple flow speeds through the analyser and not only maximal instrument flow? Increasing instrument attenuation complicates your setup not helping to answer your research question.

Valid point, corrected in new experiments

L. 108    Indicate in this section if any calibrations were performed and if not, why. This should also clarify whether the isotopic compositions you mention are raw analyser outputs or independent compositions.

Corrected in Sect. 2.3.

L. 124    If possible, indicate a range instead of the "aproximately" as the consistency of the 100 foot length seems essential for your tests! E.g. +/- 10 foot or 0.1 foot?

Now listed in Supplementary Table S1

L. 134    Clarify how the temperature was measured and guaranteed? In case 60C was the maximum heating temperature of the self-regulating heating tape, real temperatures could have been much lower (often a linear wattage decrease with no heat emitted at 60C, and hardly any at 50C. Given a heat loss through the insulation, a Win == Wout at 40C might also have been realistic).

The thermocouple probe was placed inside the insulation on the side of the tubing opposite of the heat tape, ~3 inches (~7.6 cm) from the end closest to the analyzer inlet. Differences in the insulation properties of the two materials used and likely differences in thermocouple placement relative to unavoidable gradients in temperature resulted in differences in average temperatures for each experiment, ranging from 48.6 to 75.2 °C. Specific temperatures are listed in Table S1. More precise control of temperature was not possible with the equipment in our lab. Even so, we believe the results are useful in testing performance and are likely similar to common field inlet installations.

L. 138    You mention multiple errors for the 1.1L/min flowrate. Is it 0.15, 0.45, or the combination? Relatively, errors seem large given your dependence on a consistent setup.

Omega is no longer used to determine flow rates. We instead use the Bios Definer 220.

L. 148    Explain why the filter was needed. The dry air source and dryrite are likely already filtered, and standards used were likely demineralized. It seems like an extra uncertainty that is not evidently needed.

It was a standard part of field inlet setups out of an abundance of caution and we wanted to represent that. It's removed in the new setup.

L. 209    Be more explicit about the nature of this breakpoint analysis. Is it the time from switch to any "new isotopic signal" hitting the analyser?

Yes- noted now in methods

L. 248    The contents of this paragraph are near identical to the contents of the paragraph before it in another wording, consider merging both.

We have separated results into visual and memory metric sections to prevent repetition of results.

L. 323    The "shape" of the attenuation curve was not expected to differ, but the attenuation time is expected to differ. Remove "shape" in the text to prevent this confusion for the reader.

We have kept in most mentions of "shape" of the attenuation curve to highlight that while the shape is similar, the exact coordinates of the attenuation curve and hence the attenuation times differ.

L. 327    If instrument influence is "likely" much larger than the tubing, the paper loses its merit, and the conclusions can't be made. If this is indeed a real concern this discussion point should be expanded, or experiments should be repeated.

Experiments were repeated at faster flow rate

L. 350    Unclear argument. Was the flowrate adjusted in the "with omega / without omega" experiment in the appendix? It seems like it was, making it odd to say the omega was the cause of the attenuation, and not increased analyser flow.

This is corrected in the new experiments.

L. 355    The arguments presented for MFM attenuation suggest that material type and additional volume are key, seemingly contradicting the presented conclusions stating that neither tube length (i.e. volume) nor material type impact attenuation. Please attempt to reconcile why this could be.

It could be a different material not tested. For example, the new Bev-A-Line tubing tested has a very large influence on memory. But the omega is removed in the new experiments.

L. 369    Mention this residence (or turnover) time in the methods together with your instrument and flow details. Also, the 8-12 seconds seems to be based on the 0.2-0.3 flowrates while 0.1L/min was used for most tests.

This has been added.

"This led to an analyzer residence time of 3.97 s. Calculated test tubing residence times were 1.2 s for short thick-walled FEP, 22.7 ± 0.2 s for long thick-walled tubing, and 50.8 ± 0.8 s for long thin-walled tubing." Section 2.2

L. 370    Unclear scentence. Define "this" (2x). you seem to suggest in-line elements impact the analyzers turnover time, but the analyser regulates its own inlet speed, correct?

Removed tests on in-line elements. The analyzer does regulate its own inlet speed. The MFM was influencing the isotopic attenuation time (or memory effect).

L. 388    It is not entirely clear how you defined your lag time using the breakpoints. The unexplained lag time is similar to the order of magnitude of the sample cell residence time I found for 0.1l/min flow (25s).

We have included a clearer explanation of how breakpoint analysis is conducted by looking for a change in slope of the curve (Sect. 2.4.1). A section comparing lag and new location times extracted from an impulse response analysis has also been included (Sect. 3.1).

L. 445    The recommendation to use short inlet tubing seems to contradict your own findings that inlet tube length does not matter for isotope attenuation times. Clarify that recommendation if you chose to keep it.

We have clarified recommendations based on the results of our new experiments.

L. 475    Your experiment with reduced tube ID effectively increased the air flow rate in the test tube and you found no difference in attenuation whatsoever. Stating in your conclusion that "higher air flow rates will minimize the memory effect" seems opposed to this. Please explain this better earlier in your text or adjust the sentence.

We have clarified recommendations based on results and other previous work. It is clear that the higher flow through the analyzer is important.

L. 482    To keep the conclusion short, remove "While differences…" as this sentence has no different message than the sentence before.

We have removed repetition.
* * *
**Community comments**

Review of the manuscript entitled "Water vapor stable isotope memory effects of common tubing materials"

This article compares the memory effect lag times associated with laser-based water isotopic measurements for different tubing materials and dimensions and will be very useful for people involved in laser-based water isotopic measurements. Since water isotopes in vapor are a

common measurement in atmospheric sciences, and dealing with the memory effect is a major analytical challenge, this manuscript is suited for the scope of the journal.

My general impression is that this is a good paper with important conclusions. Flow rate is more important than material being used and that the optical cavity of the system which generally cannot be modified is the largest contributor of the memory effect. This is crucial information.

**Major comments**

I would add more literature describing analytical methods used to reduce memory effect other than ValLet-Coulomb et al. 2021 such as (Guidotti et al., 2013; Schauer et al., 2016; Pierchala et al., 2019; Qu et al., 2020; de Graaf et al., 2020; Hachgenei et al., 2022)

Edited to include de Graaf and Hachgenei. This is just an example list and not meant to be an exhaustive list of studies on reducing memory influence in liquid water injection systems.

The use of feet and inches is confusing. I would remain with the metric system.

Edited to include both imperial and metric units.

Figure 3 shows that heated tubing has longer attenuation times (effectively similar within measurement error but still slightly longer). This should be addressed. It is counterintuitive – I would expect higher temperature to decrease the attenuation time. I would also suggest trying an even warmer temperature like 90 degrees.

Obtaining the equipment to do higher temperature experiments was outside the scope of this project. We also thought that higher temperatures would decrease the memory times. The new experiments now show attenuation times are shorter in the heated tubing, in general. The old experiments didn't have the resolution needed to see these slight changes in attenuation times. We attempt to address this in Sect. 3 and 4.

**Minor comments**

Line 60-61: the authors cite references claiming air flow rates and temperatures affect attenuation times, yet their results do not replicate this. I think this should be discussed

In the previous experiments, the analyzer memory was sufficiently large to limit the resolution of tubing differences. We repeated the experiments using faster flow through the analyzer and slower flow through the tubing and found slight effects of temperature and effective flow velocities. See revised Section 3.

Line 80: This definition is not accurate. The physical reason for the ME is that water molecules adsorb onto surfaces due to hydrogen bonding, which is a well-known phenomenon in vacuum technology. Replacing ordinary hydrogen with deuterium increases binding energy and, consequently, also the residence time of deuterated water molecules on internal surfaces of vacuum systems. This is why the memory effect is stronger for $\delta D$ compared to $\delta^{18}O$ as stated in line 81. The delay in the speed at which the isotopologues move through the tubing" is relevant only for diffusive transport, not for air flow.

Thank you for pointing this out- we have changed the wording.

Line 120: can delete "following Rayleigh fractionation".

We have removed this phrase.

Line 134: Specify how the temperature was measured and guaranteed. If this is the temperature of the heating tape the real temperature could have been much lower within the tubing. Especially for different flow speeds. I am concerned the fast flow rate experiment was at a lower temperature than the slow flow rate.

All flow rates through the tubing were the same. Flow should not bias tubing temperature. Different insulations do bias tubing temperature though and some of this variability is noted in the supplementary table S1 and text

"Differences in the insulation properties of the two materials used and likely differences in thermocouple placement relative to unavoidable gradients in temperature resulted in differences in average temperatures for each experiment, ranging from 48.6 to 75.2 °C (Table S1). All heated experiments (average 60 ± 8 °C) are significantly warmer than ambient temperature experiments (average 24 ±1 °C). "

Line 161: why normalize? I think there should also be comparisons for different isotopic values

We normalize to compare across experiments and avoid the need to adjust for source water and analyzer drift over time. The isotopic differences between experiments were

155.43 ± 3.43 ‰ (~2.2% variance) $\delta D$, 19.839 ± 0.463 ‰ (~2.3 % variance) $\delta 18O$, and 4.04 ± 3.610 ‰ (~89% variance) D-excess.

Line 248- 260: This paragraph can be significantly shortened to be more comprehensible. For instance L251: "the measured $t_{95\%}$ values for $\delta^{18}O$ range from 33–34 seconds, with uncertainties ranging from 32–36 seconds" can be shortened to "the measured $t_{95\%}$ values for $\delta^{18}O$ are 33-34±2 seconds"

We use this presentation in the new manuscript.

L255: either "from... to" or "between... and". Don't use "from... and"

Adjusted when describing ranges of values.

L260: Why is a second attenuation metric for judging D-excess not appropriate? This sentence is unclear.

The second attenuation time (similar to that of $\delta^{18}O$ and $\delta D$) for D-excess is not appropriate as the shape of the attenuation curve is not a unidirectional transition. We chose a time to achieve within 3 per mil based on common analyzer precision metrics. We note this more clearly in Sect. 2.4.1.

L327: Why does this disprove that the analyzer optical cavity and internal plumbing which are likely much larger than tubing effects is the dominant factor being more significant than tubing type and dimensions?

We agree with the reviewer's concerns and repeated the experiments.

L329: "the exchange rate of water molecules from the vapor to the inner tubing surface can be considered a first-order kinetic reaction". Also the exchange rate of water molecules from the vapor to the analyzer cavity and internal plumbing. In this case the internal cavity is much warmer than the tubing and may have different rate of water adsorption/desorption.

This is a true statement. However, we have removed references to reaction kinetics in this round of the manuscript.

L445: If the tubing length is not an issue according to your findings why is the recommendation to use short inlet tubing logical? It seems to contradict your findings that inlet tube length does not matter for isotope attenuation times. This should be clarified.

We have clarified recommendations based on new results and other previous work.

L495: The given link to figure data, scripts and workup code doesn't work

Fixed

**Citation(of review)**: https://doi.org/10.5194/amt-2023-56-CC1

---

## Referee Report (RR1)

**Water vapor stable isotope memory effects of common tubing materials**

Alexandra L. Meyer[1], Lisa R. Welp[1]

[1]Department of Earth, Atmospheric, and Planetary Sciences, Purdue University, West Lafayette, 47907, United States

*Correspondence to: almeyer269@gmail.com or lwelp@purdue.edu*

**Summary of Comments on amt-2023-56-manuscript-version3.pdf**

Thanks for submitting and improving opon this manuscript, which details a set of experiments aimed to find the differences between the memory effects of various tubing materials. The work is original and valuable for a broad community of research groups performing isotopic composition measurements.

The manuscript has been improved significantly compared to the previous submission. It feels more complete, it reads better, and most importantly uses a clearer/simpler experimental setup. All of my comments (find them in the document below) are minor comments related to minor logical issues or grammatical issues, with the exception of one major comment related to the inclusion of the Bev-A-Line tubing as a 'not to use' example. The residence times derived for this tubing material are simply unrealistic. I cannot believe it is possible that this tubing type, which is made of PE on the inside - Note; the same material then the HDPE tubing which is also tested - has a unrecognizebly different attenuation behavior then all other tubing materials tested. If the authors are really confident and comfortable in presenting this as a result, I would expect a strong discussion on the reasons for this outlying behavior. However, in the current manuscript, the discussion of dielectric constants and other physical features merely treat the 5 other tubing types which performed practically identically. Moreover, the supplement suggests that many of the metrics derived by the analyzers are way of for this tubing material compared to all other experiments with other tubing materials. For one, there hardly seems to be any difference between the two water vapor composition source streams in terms of isotopic compositions. (when feeding source water through whatever tube for long enough, the water coming out of the tube should become identical in isotopic to the water going in..)

I suspect that something went wrong during the experiment with the Bev-A-line, which causes the unlikely results. A repeat of only this experiment, which confirms or denies the observed behavior would be my strong suggestion before the work is published. When doing one such more test, I would suggest redoing one of the well behaving plastic tubings to confirm that the system is indeed returning results like expected. Finally, if this time around there is Decabon availible, I would also recommend including it as a true 'known not to use' example to put extra strength to your potential re-observation that Bev-A-Line has totally diverging behavior.

*Please find my line by line comments below. Note that the comments are based on an acrobat highlight markup, where some highlights (especially the green ones) do not have a connected comment.*

[revised manuscript text omitted]

Author:          Subject: Highlight          Date: 11/12/2023, 14:22:32

Author:          Subject: Highlight          Date: 11/12/2023, 14:29:04
Clear, possibly mention/refer to this when specifying the specs of the standards too.

Author:          Subject: Highlight          Date: 11/12/2023, 14:29:28
made with or taken by?

Author:          Subject: Highlight          Date: 12/12/2023, 08:34:24
This deserves a less mind braking explanation

Author:          Subject: Highlight          Date: 12/12/2023, 08:35:52
idem

mean was applied, and the 5 replicates were averaged. Replicates were screened based on successful WVISS-to-DPG and DPG-to-WVISS switching and consistent water vapor mixing ratios ensuring that vapor source generators were operating properly. Only one replicate was discarded from the heated PFA experiment due to water mixing ratio variability from the WVISS. We calculated the average D-excess value over 600–1200 seconds after the source switch and subtracted that value from all data points to adjust for small changes in D-excess source waters between replicates, especially in the DPG vapor which undergoes evaporative enrichment. The 600–1200 seconds after the source switch visually appear to be conditions of tubing equilibration and were used to calculate source vapor sample averages given in Table S1 and summarized in Sect. 2.2.

When comparing experiments between different tubing lengths and IDs, differences in the internal volume result in different tubing residence times due to advection. The flow in all experiments was laminar with Reynold's numbers calculated between 579 and 870. In Sect. 3.1 we describe how the experiments are delay-adjusted to compare transitions directly.

Memory analysis included both directions of the isotopic switch. Isotopically enriched-to-depleted (WVISS-to-DPG) figures are presented in the main body of the text, and isotopically depleted-to-enriched (DPG-to-WVISS) transitions are available in the supplemental information (Fig. S3 and S4). While Aemisegger et al. (2012) found the enriched-to-depleted switch exhibited longer attenuation times, this was likely due to the change in water vapor mixing ratio of the sources in their experiment which did not occur here.

**2.4 Memory Quantification**

Memory effects are analogous to a low-pass filter (e.g. Zannoni et al., 2022). Previous studies have approximated the smoothing of a step-change input as an exponential transition and report a threshold time to some percentage of completion like an e-folding (63 %), 90 %, or 95 % (Sturm and Knohl, 2010; Schmidt et al., 2010; Aemisegger et al., 2012; Steen-Larsen et al., 2014). In some cases, the threshold metrics were obtained from the data directly (Sturm and Knohl, 2010; Steen-Larsen et al., 2014) and in others it appears an exponential function was fit to the data first and the metrics were extracted from the fit (Schmidt et al., 2010; Aemisegger et al., 2012). A second method used in the literature takes the first derivative of the normalized transition (Steen-Larsen et al., 2014) and characterizes an impulse response function using curve fitting (Jones et al., 2017; Kahle et al., 2018). We have quantified memory effect metrics using both methods.

**2.4.1 Threshold metrics**

We extracted attenuation threshold metrics directly from the normalized and replicate-averaged data (not an exponential fit). An e-folding time corresponds to $\tau = 1/e$ of the signal transition remaining to reach a new value. In this study, we have chosen to estimate attenuation threshold times at $1\tau$ (~63 %) and $3\tau$ (~95 %) completion of the switch to the next $\delta D$ and $\delta^{18}O$ value, denoted as $t_{63\%}$ or $t_{95\%}$ respectively (Schmidt et al., 2010). These $t$ values are the time the averaged curve intersects the threshold percent value. We chose not to fit exponential curves to extract an e-folding time, because the measured attenuation curves were not accurately described by an exponential curve (not shown). The 1 standard deviation envelope was calculated by taking the standard deviation of the 5 replicates at

Author: Subject: Highlight Date: 15/12/2023, 17:24:32

Author: Subject: Highlight Date: 15/12/2023, 17:26:35
I agree partially. The effect smooths the high frequency pertubations, but also introduces a literal lag time, right?

[revised manuscript text omitted]

Author:    Subject: Highlight    Date: 18/12/2023, 15:01:23
Again, this is extremely implausible.

Author:    Subject: Highlight    Date: 18/12/2023, 15:08:59
Conclusion / discussion?

[Figure]

**Figure 3.** Mean attenuation curves for enriched-to-depleted (WVISS-to-DPG) transitions of five replicates of each tubing type for $\delta^{18}O$ (a, b), $\delta D$ (c, d), and D-excess (e, f) plotted as location-adjusted time since source switch. The first column (panels a, c, and e) depict time from -5 to 100 s, while the second column (panels b, d, and f) depicts time from -5 to 15 s. Solid lines indicate unheated experiments, while dashed lines indicate heated experiments. An orange curve in panel b shows mean $\delta D$ for comparison and in panel d shows $\delta^{18}O$ for comparison. To compensate for small differences in isotopic values between experiments, $\delta D$ and $\delta^{18}O$ are normalized from 1–0 with one at equilibrium with the first vapor source and zero at equilibrium with the second vapor source. D-excess is adjusted to end at 0 ‰ for each experiment. Gray horizontal lines indicate thresholds of 95 % and 63 % transition completion for $\delta D$ and $\delta^{18}O$, and 3 ‰ for D-excess, while a black line indicates 100 % equilibrium completion for all isotopes. Bev-A-Line XX is shown in panels a and c as a black line and never reaches a normalized 0 or 1 when compared to the experiment immediately prior. Depleted-to-enriched results are presented in the supplemental, as there were no consistent and large differences in attenuation curves between source switching directions.

When testing differences in tubing temperature and dimensions using the same material, properties affecting transit time through the tubing, like tubing length, inner diameter, and effective flow velocities, do not appear to greatly influence the shape of the attenuation curve after location adjustment (Fig. 4 and S4). The short and long thick-walled tubing $\delta^{18}O$ and $\delta D$ signals overlap each other (Fig. 4b and d), while the long thin-walled tubing has a shallower $\delta^{18}O$ slope (Fig. 4b) and a bigger delay between the $\delta D$ and $\delta^{18}O$ signal transitions (Fig. 4d). Because we've effectively normalized for tubing length, volume, and temperature through the $\delta^{18}O$ location adjustment, any

**Page: 13**

You indicate that you scale each 0 and 1 to the beginning and ending values of the transition curves. This is clear for all tubing types, but for bev-a-line it is clear that neighter 0 or 1 is ever reached. How can this be if you rescale?

More importantly, appendix table S1 indicates that the averaged isotopic compositions before and after are simply uncomparable to all other experiments. It cannot be possible that this PE tube is so porous that you cannot measure the isotopic composition of a constant source through it after equilibrating for hours.

differences in the attenuation curve steepness could be attributed to vapor-wall interactions that are independent of bulk flow.

[Figure]

**Figure 4.** Mean attenuation curves for only FEP tubing for enriched-to-depleted (WVISS-to-DPG) transitions comparing tubing length and inner diameter for $\delta^{18}$O (a, b), $\delta$D (c, d), and D-excess (e, f) plotted as location adjusted time since source switch. The first column (panels a, c, and e) depicts time from -5 to 100 s, while the second column (panels b, d, and f) depicts time from -5 to 15 s. Solid lines indicate unheated experiments, while dashed lines indicate heated experiments. To compensate for small differences in isotopic values between experiments, $\delta$D and $\delta^{18}$O are normalized from 1–0 with one at equilibrium with the first vapor source and zero at equilibrium with the second vapor source, and D-excess is adjusted to end at 0 ‰ for each experiment. Gray horizontal lines indicate thresholds of 95 % and 63 % transition completion for $\delta$D and $\delta^{18}$O, and 3 ‰ for D-excess, while a black line indicates 100 % completion for all isotopes. The location adjustment for the short tubing is much shorter than that of the long tubing, leading to a line that appears to start abruptly at ~ -3 s.

**3.3 Quantitative memory metrics**

Quantitative metrics of $\sigma_s$, $\sigma_m$, $t_{95\%}$ and $t_{63\%}$ for $\delta$D and $\delta^{18}$O, or $t_{3‰}$ and absolute value of the maximum peak for D-excess were also used to compare tubing experiments (Table S2). The different memory metrics calculated provide a

different order of "best" to "worst" tubing materials and conditions based on slight differences, though all tubings appear operationally similar (Table S2). However, some common patterns emerge. According to most impulse response metrics ($\sigma_s$ and $\sigma_m$), short thick-walled FEP has the fastest attenuation impulse response time. The slowest attenuation impulse response time for $\delta D$ is consistently found in the long thin-walled FEP, while for $\delta^{18}O$ the

410     slowest attenuation impulse response times are found in unheated copper ($\sigma_m$, enriched-to-depleted), unheated PFA ($\sigma_s$, enriched-to-depleted), and heated PTFE (both metrics, depleted-to-enriched). In terms of residence time adjusted $t_{63\%}$ values, unheated copper is the worst and short thick-walled FEP is the best for both $\delta^{18}O$ and $\delta D$. Similarly, $\delta^{18}O$ residence time adjusted $t_{95\%}$ values are longest for unheated copper and shortest for short thick-walled FEP in both directions of the isotopic switch. For residence time adjusted $\delta D$ $t_{95\%}$ times, long thin-walled FEP is the worst

415     in the enriched-to-depleted direction while heated PTFE is the worst in the depleted-to-enriched direction. Short thick-walled FEP is the best in terms of $\delta D$ residence time adjusted $t_{95\%}$ time. Short thick-walled FEP was consistently the best for $t_{3‰}$ and the absolute value of the maximum D-excess peak values, while heated long thin-walled FEP was the worse in both metrics in the enriched-to-depleted switch. In the depleted-to-enriched switch direction, heated PTFE was worse for $t_{3‰}$ but for the absolute value of the maximum D-excess peak value, heated

420     long thin-walled FEP was the worst. The rest of the tubing material types vary in their ranking depending on the memory metric used. Overall, heated memory metrics are generally either similar to or faster than those of the unheated memory metrics when comparing the same tubing types (Fig. S1d). However, this pattern does not hold for $\delta D$ $t_{95\%}$, with differences of up to 15 s between heated and unheated PTFE, with unheated signal equilibrating faster.

    Residence time adjusted attenuation threshold times are somewhat consistent with the visual analysis of Fig. 3,

425     4, S3, and S4. The residence time adjusted $t_{95\%}$ values for $\delta^{18}O$ range from 6.9–22.8 seconds with an uncertainty of up to 24 seconds for individual $t_{95\%}$ values. Measured values of $t_{95\%}$ for $\delta D$ range from 6.9–48 seconds, with uncertainties of up to 14 seconds. Because of the shallow slope of the attenuation curves at $t_{95\%}$ values contributing to large error estimates, we also report $t_{63\%}$ values because they have smaller uncertainty estimates and may have a different sensitivity to tubing differences. For our analyzer settings, residence time adjusted $t_{63\%}$ values range from

430     approximately 4.9–17.8 s for both $\delta^{18}O$ and $\delta D$, with uncertainty on the order of one second. $t_{63\%}$ values are more similar between $\delta^{18}O$ and $\delta D$ than $t_{95\%}$ values (Fig. S1d). Finally, residence time adjusted $t_{3‰}$ values for D-excess range from 0–93 seconds, while the largest $t_{3‰}$ uncertainty value was 536 seconds. D-excess $t_{3‰}$ values overlap both $\delta D$ and $\delta^{18}O$ $t_{95\%}$ ranges. We also measured the absolute value of the maximum D-excess peak or the magnitude of the transient anomaly in D-excess signals. These values ranged from ~0–31‰, inclusive of error. The average

435     difference between the beginning and ending D-excess values was 4.0 ‰.

    In terms of the impulse response method, $\sigma_m$ values which characterize the longer memory tail of the impulse function were on average, longer for $\delta D$ than $\delta^{18}O$ and ranged from 0.66–2.2 $\pm$ 0.02 s (Table S2). Mixing times ($\sigma_s$) from the skew-normal impulse function fit ranged from 1.4–5.9 $\pm$ 1.2 s and were also on average, longer for $\delta D$ than $\delta^{18}O$ (Table S2). Overall, impulse response metrics varied as expected for $\delta D$ with length and volume with longer

440     memory times for longer and larger volume tubing, but were inconsistent in $\delta^{18}O$. We were unable to calculate impulse response metrics for Bev-A-Line XX, as the isotopic switch was not achieved within the hour-long source switching.

Author:     Subject: Highlight     Date: 18/12/2023, 16:03:29
repeated use of word 'different'. Also, the message of the sentence is unclear.

Author:     Subject: Highlight     Date: 18/12/2023, 16:04:30
Strange use of However, after a scentence that already has an 'however' structure!

Author:     Subject: Highlight     Date: 18/12/2023, 16:19:33

Author:     Subject: Highlight     Date: 18/12/2023, 16:36:49
Where are the D-excess curves for BEV-A-Line?

Author:     Subject: Highlight     Date: 18/12/2023, 16:56:59
This is not possible. All the D-excess t3 trashhold values occur after t=5 seconds. Possibly your algorithm finds a zero crossing instead of a zero crossing with a negative slope.

Author:     Subject: Highlight     Date: 18/12/2023, 17:00:19
make a proper scentence.

Author:     Subject: Highlight     Date: 18/12/2023, 17:01:38

**3.4 Review of material properties**

Predictions of tubing material performance can be made based on material properties. Hydrophobic materials that are nonpolar and have a high relative permittivity (also known as the dielectric constant, or a material's ability to prevent electrical fields from forming) are ideal for water vapor isotope studies as polar water molecules are affected by and can induce electric fields (Aemisegger et al., 2012). As previously shown, $\delta D$ signal transitions are slowed compared to $\delta^{18}O$ signals, due to isotope-dependent hydrogen-bonding interactions with tubing walls. Limiting these interactions should lead to reduced isotopic attenuation times. Material specifications vary by manufacturer and material purity, but in general, FEP and PTFE materials are expected to have the least amount of water absorption of the tubing types we tested (Table 2). Metals have a relative permittivity value of ~1 due to their sea of electrons, which in this case interact with the polar water molecules. Larger values of relative permittivity are better in this case, as water vapor molecules will be less attracted to the material. HDPE, FEP and PTFE have the highest ability to prevent electrical fields. FEP and PTFE may be expected to have the shortest isotopic attenuation times based on combined water absorption percentage and relative permittivity. However, at the air flow rate we tested, the memory metrics of FEP and PTFE were not noticeably superior to the other tubing tested.

**Table 2.** Material properties of tubing type options and their water absorption percentages and relative permittivity values.

| Material | Water absorption % by tubing weight | Relative Permittivity (Dielectric constant) @ 1 MHz ($\varepsilon_r$) |
|---|---|---|
| FEP | <0.01[1] | 2.1[2] |
| PFA | <0.03[1] | 2.05–2.06[2] |
| PTFE | <0.01[1] | 2.0–2.1[2] |
| HDPE | 0.10[1] | 2.3–2.4[2] |
| Copper | N/A | ~1 |

[1] after being submerged for 24 hours (ASTM D570). This metric is solely for plastic materials    [2] (Electrical properties of plastic materials, 2021)

**4 Discussion**

Previous water vapor isotope studies have tried to identify suitable tubing material to use in sample inlets, and authors found several materials to be acceptable. To our knowledge, these materials had not be rigorously tested for wall adsorption/desorption effects leading to memory artifacts. Theory based on principles of gas chromatography and gas-wall partitioning predicts that the residence time of gases adsorbed on tubing walls is linearly proportional to tubing inner diameter and should decrease at higher temperatures as gas saturation concentration changes (Pagonis et al., 2017). The experiments performed in this study begin to test these predictions for water vapor isotopes.

Author:     Subject: Highlight     Date: 18/12/2023, 17:07:00
I suggest putting this subchapter under the discussion, as its not your findings that are discussed but the findings of other researchers.

Author:     Subject: Highlight     Date: 18/12/2023, 17:02:46

Author:     Subject: Highlight     Date: 18/12/2023, 17:05:34
If this is the reason we observe the difference, why does the attenuation time in no way scale with the dielectric constant?

Author:     Subject: Highlight     Date: 18/12/2023, 17:08:21
When evaluating material types like you do here, why not mention Bev-A-line and try to explain why it behaves so strangely? It is effectively PE, so the same dielectric properties HDPE are likely found.. This is a major missing point.

Author:     Subject: Highlight     Date: 18/12/2023, 17:11:13
include Bev-A-line!

**4.1 Effects of material and temperature**

We found δD and δ18O attenuation curves between tubing materials were slightly different, but operationally similar, at the flow rate, humidity, and temperatures tested (Fig. 3, 4, S3, and S4), with the exception of Bev-A-Line XX. Our results are consistent with Griffis et al.'s (2010) assertion that HDPE is similar to PTFE. Similarly, Aemisegger et al., (2012) found little difference in attenuation times with varying PFA tubing temperatures. We were not able to replicate Steen-Larsen et al.'s (2014) finding that copper was better than PTFE. In our study, tubing materials performed similarly when comparing all memory metrics: $\sigma_s$, $\sigma_m$, $t_{63\%}$, $t_{95\%}$, $t_{3‰}$, and the absolute value of the maximum D-excess peak. Variations in reported material properties presented in Sect. 3.4 predict only slight differences in gas-wall effects in the commonly used tubing materials but were unable to explain the relative differences in memory metrics measured in these conditions. We believe the differences are too small to accurately measure in this experimental setup, partially based on the additional ~4 s residence time of the analyzer optical cell and internal plumbing.

Warmer temperatures are theoretically predicted to reduce attenuation times (Pagonis et al., 2017) by changing the saturation concentration of gases. The lower molar density of the warmer air means there is a shorter residence time through the tubing, increased molecular movement, faster wall exchanges, and fewer molecules stuck to the tubing walls. We found some evidence of this in comparing location times and $\sigma_m$ from the impulse function method (Table S2). Location times for heated tubings are always faster than their unheated counterparts, and $\sigma_m$ values are similar to or shorter for heated tubings in most cases. Calculated residence times and observed lag times were also faster for heated tubings, but to varying degrees depending on the tubing. The heated tubing likely has faster residence, lag, and location times due to the decreased number of molecules in the tubing compared to the unheated experiment and possibly also due to decreased wall effects.

Tubing residence time predictions are up to 12 s shorter than the measured breakpoint lag. Uncertainties in tubing residence time (a few seconds), length (a couple inches), and breakpoint lag (a few seconds) account for some of these differences. Tubing temperature measurements in the heated treatment varied depending on the position of the thermocouple relative to the heat cable. It is expected that the tubing was not at a perfectly uniform temperature, but we note that this heating design is commonly used in field conditions and represents likely inlet conditions. However, the lack of uniform temperature control leads to potential temperature-induced differences that are hard to quantify. This should be considered when comparing residence time adjusted memory metrics between experiments. Differences not attributed to variations in temperature, length, or error in the breakpoint lag may be due to wall effects.

**4.2 Effects of tubing inner volume and length**

The model in Pagonis et al. (2017) indicates that tubing residence time is expected to scale based on length, but should not affect attenuation times sensitive to wall effects. The difference in length in the thick-walled FEP long and short experiments was a factor of 19 (99 ft/5.2 ft, or 30.2 m/1.6 m) which results in the same factor difference in residence time calculations. The breakpoint lag differences between long and short thick-walled FEP tubing was

**Page: 17**

Why are only slight differences predicted based on material? Copper is vastly diffident that the plastics used in terms of properties. If theres properties were related to the ultimate attenuation you find, this shouls be seen right?

I do not agree this can be concluded. Material properties dit differ, and 2x 20s is significantly more then 4s...

but not the majoritie of the difference right? could tube roughness play a role here?

What are these wall effects governed by if not temperature variations, temperature, or material properties?

505 approximately a factor of 8 times faster in the short tubing experiment. While there are slight differences in these memory metrics, this is likely due to the influence of the analyzer. Because the analyzer optical cavity and inner tubing has a residence time of ~ 4 sec, we are unable to resolve the residence time and memory metrics associated with the short FEP tubing (1.0 ± 0.09 s) only. Even with the large length difference, the shape of the isotopic attenuation curves remained similar after location adjustment which removes the length-based residence time

510 differences (Fig. 4 and S4). Residence time adjusted δD $t_{95\%}$ and $t_{63\%}$ times for long thick-walled FEP tubing were at maximum 3.2x and 1.6x greater than the short, respectively. The mixing time scales ($\sigma_s$) and the memory tail metric ($\sigma_m$) both showed less than a doubling between short and long tubing. These modest differences in wall-effect memory metrics are not explained by the theory in Pagonis et al. (2017).

The tubing ID affects the residence, lag, and location times, the surface area to air volume ratios, and the flow

515 velocity past the surface of the tubing interior. Pagonis et al. (2017) predicts the residence time of gas molecules on or in the tubing walls changes linearly with respect to tubing ID when the tubing material does not change. In our experiments, ID increased by a factor of 1.5x between thick- and thin-walled FEP ($^1/_8$ in. or ~3.18 mm ID compared to $^3/_{16}$ in. or ~4.76 mm ID). Separation in Fig. 4c and 4d between thick- and thin-walled FEP is exaggerated by the $\delta^{18}O$ location adjustment applied to the δD signal, but thin-walled FEP does have a slightly less steep slope and

520 longer $t_{63\%}$ intercept than the thick-walled tubing. Residence time adjusted memory metrics also show a slight increase with ID increase, with an average 1.9x larger memory metric for δD and 1.66x larger memory metric for $\delta^{18}O$ between thin- and thick-walled tubing (Table S2). The long thin-walled FEP consistently showed the slowest δD signal transitions of the tubings tested (Fig. 3, 4, S3, and S4 panels c and d). From the location adjusted comparison of the same material (FEP) with different IDs (Fig. 4, $\delta^{18}O$ location adjusted plot), we conclude that a

525 bigger ID causes the slower memory metrics. We also note that PTFE and PFA also had the same $^3/_{16}$ in. (~4.76 mm) ID and those experiments showed a faster attenuation threshold time than FEP (Fig. 3). Therefore, the material differences and tubing ID seem to play a role in our experiment, consistent with theory that tubing ID, material density, and partitioning depth will affect the residence time of chemical compounds on or in a tubing wall (Pagonis et al., 2017).

530 In summary, we found that all tubing dimensions, including ID and length, had some effects on the threshold metrics (Fig. 4 and S4) after removing differences in residence times in signal propagation to the analyzer based on tubing inner volume and the temperature influence on molecular density. While these overall memory metric differences exist, they are small in the materials and dimensions tested, and the operational impact among commonly used ¼ in. (6.35 mm) OD tubing inlets is expected to be limited.

535 **4.3 Relative attenuation time differences between δD and $\delta^{18}O$**

δD signals have been demonstrated to take longer than $\delta^{18}O$ signals to isotopically equilibrate with tubing materials due to isotope-dependent hydrogen-bonding interactions with the tubing walls (Sturm and Knohl, 2010; Griffis et al., 2010; Schmidt et al., 2010). This speed difference has been reported as a ratio of attenuation times between the slower δD signal and the faster $\delta^{18}O$ signal, and a large range of ratios have been reported. Published results show

540 1.4–3.5x greater attenuation time for δD signals than $\delta^{18}O$ signals depending on tubing air flow rate, tubing type,

**Page: 18**

Strange thing here is that decreased ID increases the relative surface area but decreases the absolute surface area.

The metrics are not slow. decreased memory times?

repeat of also

Time is not fast, but short of long.. please improve such logical errors to improve the overall readability!

Please rewrite every point where Bev-A-line is excuded from the analysis / conclusions without proper reasoning. How come that Bev-A-Line very strongly does not follow this expected behaviour?

[revised manuscript text omitted]

Author:     Subject: Highlight          Date: 21/12/2023, 13:37:36
I here mis the obseravtion that the center point of a pulse will be timedelayed w.r.t., for example, the molefraction variation of H2O itself. So this attenuation imposes both a lag time as a smoothing effect.

Author:     Subject: Highlight          Date: 21/12/2023, 13:41:40

Author:     Subject: Highlight          Date: 21/12/2023, 13:41:55

data was collected at 1 Hz, their reported data is a 15 s average, which allows them a 975 m horizonal and 75 m vertical resolution (Sodemann et al., 2017). However, that best-case estimate is based on the data averaging interval and does not consider signal attenuation due to tubing isotopic memory or mixing in the optical cavity (Sodemann et al., 2017). Additionally, averaging over long time periods may not remove D-excess memory bias depending on patterns of increasing or decreasing delta values. The wetting and drying of the measurement system during flights with large changes in altitude, and therefore atmospheric specific humidity, may also increase isotopic attenuation times but were not quantified here.

In both eddy covariance and aircraft measurement situations, one might consider increasing air flow through the analyzer and intake tubing and shortening the length of tubing from an intake pickoff point to the analyzer in slow analyzer flow setups as has been suggested in previous studies (e.g. Griffis et al., 2010). While high air flow rates can easily be achieved in the air intake main lines in both high-frequency measurement situations, the air flow rate through the analyzer is typically limited by the analyzer design and control software. If tubing or in-line elements like mass flow controllers affect the speed at which the isotopes are transmitted from the intake to the optical cavity, signals are effectively low-pass filtered (Zannoni et al., 2022). Our experiments show shorter memory effects for shorter tubing compared to longer tubing. Therefore, it is also important to minimize the length of tubing from the intake pickoff point to the analyzer to reduce the residence time of air in the low-flow portion of the system. These considerations should also maximize D-excess data resolution.

**5.3 Liquid water measurements**

Liquid water isotope analysis is also plagued by memory effects when samples are converted to the vapor phase for spectral isotopic analysis, especially in applications measuring samples with large isotopic differences in the same batch. Common protocols recommend multiple replicate injections and discarding the first few to remove carryover from the previous sample (IAEA, 2009; Penna et al., 2012; Coplen and Wassenaar, 2015). In both OA-ICOS and cavity ring-down spectroscopy, Penna et al. (2012) found that when measuring samples with large isotopic differences, up to eight out of eighteen injections had to be ignored to limit memory effects. When analyzing highly depleted Antarctic samples ranging from −231.7 ‰ to −421.1 ‰ for δD, memory effects of up to 14 ‰ were found in the first injection compared to the "true" value. Liquid water analysis is one example of a case where air flow rates and temperatures of transfer lines are fixed by the instrument design. Material properties inside the analyzer are important, but this study finds little difference between commonly used material types. Waiting for equilibrium in the optical cavity may minimize the memory effect, but a time-efficient method to increase sample throughput is to mathematically correct for these repeatable effects rather than attempting to minimize them (e.g. de Graaf et al., 2020; Vallet-Coulomb et al., 2021; Hachgenei et al., 2022). Or, in the case of de Graaf et al., (2020), one can measure small vapor samples on a background of humid air to reduce memory effects. Work is also being done in the ice core community to correct out signal mixing based on transfer function fitting methods (e.g Jones et al., 2017; Kahle et al., 2018). These memory correction approaches may provide examples of methods to reconstruct input signal variability from smoothed continuous vapor isotope measurements as well.

Author:          Subject: Highlight          Date: 21/12/2023, 13:42:54

Author:          Subject: Highlight          Date: 21/12/2023, 13:43:38

Author:          Subject: Highlight          Date: 21/12/2023, 13:44:36
Spectral? with laser spectrometers you mean? Mass spec is also spectal.

Author:          Subject: Highlight          Date: 21/12/2023, 13:46:32
While this is certainly often the case, it is not nececarally true.

**6 Conclusions**

We tested the water isotopic exchange properties of PFA, FEP, PTFE, HDPE, copper, and Bev-A-Line XX. The commonly used materials tested here perform similarly. It does not seem necessary to standardize materials used to measure stable water vapor isotopologues to make accurate and comparable measurements in most situations, when using analyzers with similar residence times. We cannot recommend Bev-A-Line XX for use in water vapor applications due to extremely long attenuation times. Warmer temperatures did shorten the residence time, lag, and location metrics of the impulse function and $t_{63\%}$ threshold times across all long tubing experiments but results were not always consistent for $t_{95\%}$. While differences may be found among tubing material types at lower humidity or while changing humidity, these experiments are beyond the scope of this study. Larger tubing IDs were predicted to increase memory metrics proportionally based on gas-wall partitioning theory (Pagonis et al., 2017), and we found that tubing ID and length had some effects on the threshold metrics after removing differences in residence times. The experiments here showed overall memory metric differences do exist, but that they are small in the materials and dimensions tested. In experimental settings, operational impact among commonly used ¼ in. (6.35 mm) OD tubing inlets is expected to be limited.

Researchers must understand the limitations of the air flow conditions and wall effects of their instrumental and intake setups to limit signal memory effects, especially if low air flow rates are a constraint or if there are large isotopic variations over short periods of time. Our experience and results from other published studies indicate that maximizing air flow rates through the analyzer is the most effective way to minimize memory effects when accurate high-frequency D-excess measurements are desired. As each individual analyzer is unique, users are advised to test their analyzer for memory effects with no intake tubing. Our results show that these plastic tubing materials are not inferior to copper in terms of isotopic memory under the tested conditions, and they are easier to work with and are less expensive than copper. As with most decisions, environmental conditions, cost, and preference may influence the type of tubing selected.

**Code/Data Availability**

All figure data and scripts, as well as an example workup code, are available at https://doi.org/10.4231/T6J3-H649 (Meyer and Welp, 2023).

**Author Contributions**

ALM and LRW designed the experiments and conducted them. ALM adapted code from LRW and added to it for this project, as well as analyzed data. ALM wrote the manuscript draft. ALM and LRW edited the document.

**Competing Interests**

The authors declare that they have no conflict of interest.

**Acknowledgements**

We thank Matthew Binkley (MS Materials Engineering) for valuable discussion of material properties.

**Financial Support**

AM was supported by a Purdue Doctoral Fellowship and the National Science Foundation Graduate Research Fellowship Program under Grant No. (DGE-1333468). Any opinions, findings, and conclusions or recommendations

**Page: 22**

What does the XX specification indicate? may be good to mention somewhere!

I think it is important to mention that your RH is in any case not above 50%, even without heating the inlet. As far as I understand, the community uses heated inlets when RH~95 may occur without heating, which starts to form a risk for liquid water films.

In which direction? does this allow you to draw conclusions on the relative surface area (in m per m2 flow area) vs absolute surface area importances?

This suggests some randomness that I do not believe is supported by your data or the exchange principles you detail.

[revised manuscript text omitted]

---

## Referee Report (RR2)

I would like to thank the authors for their effort in incorporating the recommended suggestions and redoing some tests with the Bev-A-Line XX tubing, which included remeasured HDPE tubing. In addition, the inclusion of Decabon in the analysis is appreciated. Importantly, you clearly indicate that the Bev-A-Line XX tubing indeed performs as poorly as you stated previously. Please excuse my earlier scepticism on that finding.

I have few comments left on the rest of the manuscript, which I believe is near ready for publication. It will be valuable for future water isotope analysis studies to have this tubing material comparison available.

The experiments are now well explained, and the results easy to interpret. I am however left somewhat unsatisfied / hugely curious to the process driving the wall exchange in the poorly performing tubes, which currently remains an open question. I understand that you do not know the answer to that question, but some discussion specific to that topic would be interesting for the curious reader (optional).

Small comments:

L. 369; there is no table S2. Table S1? Optionally change the name of the supplement headers (S1-S6) to not match the figure and table names.

L. 390; No location time in table S1/S2.

L. 440; Try to explain how or why the Dekabon dD and d18O transitions are so different.

L. 566; While the Decabon and Bev-A-Line XX liner materials are unknown, it is worth speculating on the process causing the large smoothing/wall exchange observed. Especially the H2O panel in figure S3 begs the question of where the WVISS water physically goes (probably the liquid phase).

L. 609; amount → margin

L. 679; Try to give some indication / explanation of why such a factor of 71 can exist. What kind of process can be affecting dD so much more?

L. 795; isotopic change → the isotopic step change

Figure S4 and Figure S9 seem to have accidental double axis labels.

---

## Author Response (AR2)

Author's note to editor: We have taken the reviewer's recommendation that Dekabon be presented as the case of a poor performing tubing material for water vapor isotope measurements. The revised manuscript is based on the following major changes.

1. We added a new tubing material, Dekabon, which demonstrates a long memory effect (longer than the commonly used materials) and therefore provides an unacceptable material example. This major change allows us to remove Bev-A-Line XX, a tubing not commonly used in water isotope studies. We have decided to remove Bev-A-Line XX result presentations in the main manuscript because the mixing ratio and isotopic memories are so excessively long that full equilibration takes half a day. Quantifying the memory metrics becomes impractical using the methods designed for the commonly used materials. We would like the reviewers to know that experimental errors are not the cause of excessive memory. It really is that bad. As a service to the research community, we would like to retain a note in the main manuscript that it is not recommended for studies. Evidence, without replicates is provided in the Supplemental.

2. We repeated a subset of experiments (HDPE and Dekabon) with the sources at two different water concentrations. This shows the reviewer how all signals propagate through a "bad" tubing material and shows differences in tubing length when the tubing memory is larger than analyzer residence time. This also gives further insight into whether the direction of the isotopic switch matters, or whether the water mixing ratio change drives memory effects under these conditions. Differences in memory effects with isotopic switch direction have been identified in $H_2O$ varied experiments, but not in $H_2O$ matched experiments.

3. Differences in memory effects based on tubing length and ID have now been identified, following theory. However, we believe our results are too insensitive to make any comments on the relationships between length/ID and surface area relationships in the commonly used tubings.

Author responses are included in red text.

We appreciate the time, suggestions for improvement, and patience of editor Thomas Röckmann and the anonymous referee for this round of revision and throughout this process.

Summary of Comments on amt-2023-56-manuscriptversion3.pdf

Thanks for submitting and improving opon this manuscript, which details a set of experiments aimed to find the differences between the memory effects of various tubing materials. The work is original and valuable for a broad community of research groups performing isotopic composition measurements.

The manuscript has been improved significantly compared to the previous submission. It feels more complete, it reads better, and most importantly uses a clearer/simpler experimental setup. All of my comments (find them in the document below) are minor comments related to minor logical issues or grammatical issues, with the exception of **one major comment related to the inclusion of the Bev-A-Line tubing as a 'not to use' example. The residence times derived for this tubing material are simply unrealistic. I cannot believe it is possible that this tubing type, which is made of PE on the inside** - Note; the same material then the HDPE tubing which is also tested - has a unrecognizebly different attenuation behavior then all other tubing materials tested. If the authors are really confident and comfortable in presenting this as a result, I would expect a strong discussion on the reasons for this outlying behavior. However, in the current manuscript, the discussion of dielectric constants and other physical features merely treat the 5 other tubing types which performed practically identically. Moreover, the supplement suggests that many of the metrics derived by the analyzers are way of for this tubing material compared to all other experiments with other tubing materials. For one**, there hardly seems to be any difference between the two water vapor composition source streams in terms of isotopic compositions**. (when feeding source water through whatever tube for long enough, the water coming out of the tube should become identical in isotopic to the water going in.)

I suspect that something went wrong during the experiment with the Bev-A-line, which causes the unlikely results. A repeat of only this experiment, which confirms or denies the observed behavior would be my strong suggestion before the work is published. When doing one such more test, I would suggest **redoing one of the well behaving plastic tubings to confirm that the system is indeed returning results like expected**. Finally**, if this time around there is Decabon availible, I would also recommend including it as a true 'known not to use' example** to put extra strength to your potential re-observation that Bev- A-Line has totally diverging behavior.

Please find my line by line comments below. Note that the comments are based on an acrobat highlight markup, where some highlights (especially the green ones) do not have a connected comment.

First, we repeated the Bev-A-Line XX experiment with a HDPE control. And we added a change in water mixing ratio to further illustrate the material behavior. Bev-A-Line XX really is that terrible. Unfortunately, the material details are proprietary, and we cannot speculate on the cause of this poor performance. When adding a beaker of warm water to a coil of tubing in a closed container, the water mixing ratio increased, but we are not sure if it is because the tubing is permeable or temperature sensitive. Warming it with a heat gun also increased the water

vapor mixing ratio of the air output, indicating that it may be especially sticky for water and temperature sensitive. We considered leaving the Bev-A-Line XX results in. We are confident in the observations. However, the performance is so bad, it becomes nearly impossible to do the curve fitting in the same way as the other tubing to produce memory metrics and it requires separate figure scales to see the full equilibration time. The error of the memory metrics increases because the shape of the memory is much more stretched out. For this reason, we decided to remove the detailed Bev-A-Line XX analysis. It is plotted in the supplementary material and noted as a poor performing material.

We also want to note that Bev-A-Line XX is not the same as Bev-A-Line IV or V HT , which this reviewer may have experience with. Bev-A-Line IV and V HT indeed have a polyethylene liner, but to our knowledge Bev-A-Line XX has a patented Hytrel® liner. Attempts to find out the exact material of this Hytrel® liner proved fruitless. We have removed Bev-A-Line XX from this version of the manuscript for clarity and conciseness, but we did rerun those experiments. Full source transition was only achieved after about 6 hours.

[Figure]

**Figure S7.** Full sweepout curve of 100 foot (30.48 m) Bev-A-Line XX for $\delta^{18}O$ (a), $\delta D$ (b), D-excess (c), and $H_2O$ (d) in both depleted-to-enriched (black points) and enriched-to-depleted (red points) directions**.** Bev-A-Line XX takes approximately 6 hours to fully equilibrate in either direction of the switch. The time to equilibration is longer in the enriched-to-depleted direction.

The initial Bev-A-Line XX test from last submission was ran with 1 hour switching which was not enough time for tubing equilibration with the new vapor source. The isotopic values for each source were gathered from a short piece of FEP run immediately prior, reflecting the "true" source values that would have been achieved if the Bev-A-Line XX was allowed to equilibrate in both directions. The values reported in the supplementary table were inaccurate because full equilibrium was not achieved.

**Comments in Order (major/red comments bolded)**

L. 35 It might be good to specify that the water molecules sticking to the wall are in 'gaseous form', or at least that is what you are testing with the measurement setup.

Added.

L. 38 I get your point, but the effect is not measurement type dependent right?

Correct. Adjusted.

L. 42 new paragraph?

Adjusted.

**L. 47 This is seriously implausible.. (referring to Bev-A-Line XX results)**

Results of Bev-A-Line XX are correct. Full source transition was only achieved after about 6 hours upon retest. The setup was confirmed with a HDPE control. The reviewer is welcome to test this material themselves. But we would advise against it. See earlier comments.

**L. 51 Bev-A-line is also a commonly used plastic tubing material, isn't it?**

Bev-A-Line XX is not used in water vapor isotope studies, to our knowledge. Bev-A-Line XX is used in soil gas $O_2$ and $CO_2$ studies which is why we had it in the lab and were curious how it would perform for water vapor. Few studies in the emerging soil gas water isotope community have used Bev-A-Line IV, a different material. One could argue that we should have tested Bev-A-Line IV also, but this was outside our budget and time constraints.

L. 76 Note that Bev a line has a polyethylene liner inside. So effectively the surface molecules present i.c.w. HDPE are identical.

This is correct for Bev-A-Line IV and V HT, but as far as we can tell not for Bev-A-Line XX. The XX tubing has a patented Hytrel® liner and attempts to find out what that liner is made of were fruitless.

L. 110 Was this achieved on a synthetic air/vapor source or on ambient air? I think that is interesting for the reader

Synthetic (WVISS produced). Added.

L. 113. Allan variance curves show the entire frequency dependence of the instrument stability. The fact that you choose to display it at '2 seconds' simply means you use the variance (in this case without pre-smoothing the signal). Agree? if so, rephrase. Why you use 2 seconds is also unclear to me here. Why not choose 20 seconds, at the bottom of your 'averaging' curve. That is likely the relevant precision of your analyser.

We have clarified why we have chosen to present 2s allan deviation to quantify measurement uncertainty and refer the reader to Figure S1 to compare analyzer precision. Using a 20 sec running average would smooth away the fast transition times we are aiming to quantify and is counterproductive to this experiment objective. We include it as an estimate of analytical uncertainty.

L. 115 Again, mention if this is a measurement of ambient air or a reference measurement.

WVISS sourced. Added.

L. 120 Inform the reader about the origin of these values. Are these the measurement outputs from the LGR, and post or pre calibrations?

"measured by the LGR TWVIA without calibration" added, Sec. 2.2.1

L. 179 Clear, possibly mention/refer to this when specifying the specs of the standards too.

"measured by the analyzer without calibration" added, Sec. 2.2.1

L. 181 made with or taken by?

Made with. Adjusted.

L. 187 This deserves a less mind braking explanation

Adjusted.

L. 189 idem

Adjusted.

L. 210 I agree partially. The effect smooths the high frequency pertubations, but also introduces a literal lag time, right?

True. In this section we are following the methods for quantifying signal smoothing only. Lags are adjusted for as well in a separate section, but are not the focus of the signal smoothing discussion.

L.263 I immagine that fitting to the transfer function of Bev-a-line was also not self evident?

Correct. Fitting the transfer function was extremely difficult. Since the Bev-A-Line didn't equilibrate in either direction over 1 hour, the transfer/impulse functions were not good fits and should not have been reported in the previous manuscript draft. We have removed Bev-A-Line XX results from the main manuscript.

L.272  Why wouldn't you use the gradient in the transfer fuction itself to extract the derivative in time? Does equation 2 + additions really tell the reader something new, or would a scentence about using the gradient be identical? Please concider!

Without a citation, it's hard to determine what the reviewer is suggesting. There may be some confusion though. The transfer function is fit to the observations to remove high frequency noise which would get amplified by the derivative step. The impulse function is found by taking the first derivative of the transfer function, or d delta/dt, which is the rate of change, which could also be described by the gradient. I don't see that one is superior. Eqn 2 fits the model to the impulse function. The Eqn 2 model fit shown provides additional information in describing the shape of the impulse function following the spirit of the methods set out in Steen-Larson et al, 2014; Jones et al, 2017; and Kahle et al., 2018. So to be clear, the impulse function is the derivative (or gradient) of the transfer function, but the shape of that impulse function is examined by fitting the model (Eqn 2) and reporting the parameters that determine the shape of the impulse function.

L. 320 Could a discrepancy between theoretical and real tube roughness cause this slight difference? Note that Iassume that roughness was taken into account for calculating the flowrates here!

Flow rates were measured directly just upstream of the analyzer and not calculated. But roughness may influence the signal transition by creating more mixing at the transition front.

**L350. Again, this is extremely implausible. (referring to Bev-A-Line XX results)**

See L. 378 response

   L. 365 Conclusion / discussion? (ALM: Sentence in Sect. 3.2 is "Given differences in D-excess values between sources, we caution overinterpreting the maximum D-excess anomalies between experiments, as evidenced by the different starting points in Fig. 3e.")

L. 378 You indicate that you scale each 0 and 1 to the beginning and ending values of the transition curves. This is clear for all tubing types, but for bev-a-line it is clear that neighter 0 or 1 is ever reached. How can this be if you rescale?

More importantly, appendix table S1 indicates that the averaged isotopic compositions before and after are simply uncomparable to all other experiments. It cannot be possible that this PE tube is so porous that you cannot measure the isotopic composition of a constant source through it after equilibrating for hours.

Full source transition was only achieved after about 6 hours.

See earlier comments.

L. 406 repeated use of word 'different'. Also, the message of the sentence is unclear.

Fixed. "Each memory metric calculated provides a different ranking of tubing material based on slight numerical differences in metric values, and all tubings appear operationally similar with the exception of Dekabon (Table S2)."

L. 407 Strange use of However, after a scentence that already has an 'however' structure!

Fixed. "Some common patterns in these rankings do emerge."

L. 431 Where are the D-excess curves for BEV-A-Line?

Good catch- I forgot to add those. This has been fixed for Dekabon.

L. 432 This is not possible. All the D-excess t3 trashhold values occur after t=5 seconds. Possibly your algorithm finds a zero crossing instead of a zero crossing with a negative slope. (ALM: referencing "Finally, residence time adjusted t3‰ values for D-excess range from 0–93 seconds, while the largest t3‰ uncertainty value was 536 seconds.")

We agree that the 3 per mil threshold is reached after 5s in Figures 3, 4, and S3. The 0 seconds for D-excess threshold value was residence time adjusted unheated short thick-walled FEP (WVISS direction). In the location adjusted Figure 4s, the D-excess anomaly never exceeds 3 per mil. In this tubing and transition direction, the memory effect is so short there is basically no significant D-excess anomaly, leading to us to report a D-excess t3 value of 0 in the manuscript. However, we see the confusion, and have listed this value as N/A now in the memory metric table, and this sentence no longer appears in the manuscript.

L. 435 make a proper scentence.

Sentence removed.

L. 443 I suggest putting this subchapter under the discussion, as its not your findings that are discussed but the findings of other researchers.

Fixed. Now Sect. 4.1. We do wish to point out that we are amenable to removing this section if the editor agrees it does not add to the manuscript.

L. 451 If this is the reason we observe the difference, why does the attenuation time in no way scale with the dielectric constant?

It is possible the dielectric constant is not a good characterization of wall effects for water isotopes. See Sect 4.1 for further discussion into this topic. We do wish to point out that we are amenable to removing this section if the editor agrees it does not add to the manuscript. We were searching for predictive capability to screen tubing materials, but unfortunately, did not find that these metrics were useful.

**L. 453 When evaluating material types like you do here, why not mention Bev-A-line and try to explain why it behaves so strangely? It is effectively PE, so the same dielectric properties HDPE are likely found.. This is a major missing point.**

See previous comments for Bev-A-Line. We do not attempt to speculate on the reason material types are different- we had a hypothesis on what properties may play a role in isotopic effects but they do not appear to play a significant role based on our results.

**L. 457 include Bev-A-line! (ALM: now Dekabon)**

We cannot include Dekabon (or previously Bev-A-Line XX) due to the undisclosed patented inner liner material.

L. 477 Why are only slight differences predicted based on material? Copper is vastly diffident that the plastics used in terms of properties. If theres properties were related to the ultimate attenuation you find, this shouls be seen right?

Changed to "may predict differences". The memory effects of these tubings (with the exception of Dekabon) are very small relative to the properties of the analyzer. The properties we reference may impart non-fractionating effects, or the memory effects may be too small for these properties to have a visible impact. But we bring up % water absorption and relative permittivity as two of the material properties that seem like they would have an impact. Further research is needed into how material properties of tubing affect water isotope memory. We do wish to point out that we are amenable to removing this section if the editor agrees it does not add to the manuscript.

**L. 479 I do not agree this can be concluded. Material properties dit differ, and 2x 20s is significantly more then 4s...**

While material properties do differ, we are not attributing these differences to any operational differences in tubing performance. There are few consistent differences between tubing

temperature or material performance, and they are relatively small. However, we have removed this paragraph to limit repetition.

L. 493 but not the majoritie of the difference right? could tube roughness play a role here?

Removed paragraph to limit repetition.

L. 499 What are these wall effects governed by if not temperature variations, temperature, or material properties?

Removed paragraph to limit repetition.

L. 520 Strange thing here is that decreased ID increases the relative surface area but decreases the absolute surface area.

Thank you for this comment. It is somewhat strange, but unfortunately, we are unable to comment much further into these relationships due to the sensitivity level of our experiments.

L. 525 The metrics are not slow. decreased memory times?

Adjusted.

L. 525 repeat of also

Adjusted.

L. 526 Time is not fast, but short of long.. please improve such logical errors to improve the overall readability!

Adjusted to the best of our ability. There are certain sentences where it is not immediately clear whether "fast" or "long" would be more appropriate, so we appreciate further input on the matter from the editors.

**L. 533 Please rewrite every point where Bev-A-line is excuded from the analysis / conclusions without proper reasoning. How come that Bev-A-Line very strongly does not follow this expected behaviour?**

See previous comments for Bev-A-Line, but now considered for Dekabon.

L. 542 0.7x is not greater then. I suggest changing the wording.

Changed.

L. 577 I here mis the obseravtion that the center point of a pulse will be timedelayed w.r.t., for example, the molefraction variation of H2O itself. So this attenuation imposes both a lag time as a smoothing effect.

We now show in our results that isotopic values transition slower than the $H_2O$ signal (Section 3.4), and have adjusted the wording to include lag time and smoothing.

L. 629 Spectral? with laser spectrometers you mean? Mass spec is also spectal.

Changed to "laser-based spectral isotopic analysis"

L. 635While this is certainly often the case, it is not nececarally true.

Added "often" to sentence. Now reads "Liquid water analysis is one example of a case where air flow rates and temperatures of transfer lines are often fixed by the instrument design."

L. 647 What does the XX specification indicate? may be good to mention somewhere!

This is the manufacturer's name and is not elaborated on. As far as we can tell, the XX specification is just the numerical value added to the end of the Bev-A-Line name to distinguish it from other Bev-A-Line tubing lines, including Bev-A-Line IV and Bev-A-Line V HT.

L. 652 I think it is important to mention that your RH is in any case not above 50%, even without heating the inlet. As far as I understand, the community uses heated inlets when RH~95 may occur without heating, which starts to form a risk for liquid water films.

Added "At this relative humidity of ~33 %" and "Heating to avoid condensation does not seem to negatively impact the isotopic measurements."

L. 655 In which direction? does this allow you to draw conclusions on the relative surface area (in m per m2 flow area) vs absolute surface area importances?

Increasing tubing ID and length increased the threshold metrics. However, we have concluded that the experiments are not sensitive enough to confirm linear or non-linear behavior with surface area. Dependence on length and ID would suggest memory is a function of total surface area or number of exchange sites.

L. 664. This suggests some randomness that I do not believe is supported by your data or the exchange principles you detail.

This sentence has been cut.

References:

Steen-Larsen, H. C., Sveinbjörnsdottir, A. E., Peters, A. J., Masson-Delmotte, V., Guishard, M. P., Hsiao, G., Jouzel, J., Noone, D., Warren, J. K., and White, J. W. C.: Climatic controls on water vapor deuterium excess in the marine boundary layer of the North Atlantic based on 500 days of in situ, continuous measurements, Atmospheric Chemistry and Physics, 14, 7741–7756, https://doi.org/10.5194/acp-14-7741-2014, 2014.

Jones, T. R., White, J. W. C., Steig, E. J., Vaughn, B. H., Morris, V., Gkinis, V., Markle, B. R., and Schoenemann, S. W.: Improved methodologies for continuous-flow analysis of stable water isotopes in ice cores, Atmospheric Measurement Techniques, 10, 617–632, https://doi.org/10.5194/amt-10-617-2017, 2017.

Kahle, E. C., Holme, C., Jones, T. R., Gkinis, V., and Steig, E. J.: A Generalized Approach to Estimating Diffusion Length of Stable Water Isotopes From Ice-Core Data, Journal of Geophysical Research: Earth Surface, 123, 2377–2391, https://doi.org/10.1029/2018JF004764, 2018.

---

## Author Response (AR3)

The main comments the reviewer and editor had revolved around why/how Dekabon and/or Bev-A-Line XX performed so much worse than the rest of the other tubing materials. We have grouped our responses to address these major topics first, with the remaining comments addressed at the end of the document. Most of the edits can be found in Discussion Section 4.5.

Thank you both for your suggestions, time, and consideration throughout this process. You've made the first author's first publishing journey very pleasant!

**Major themes:**

**Line 566:** While the Decabon and Bev-A-Line XX liner materials are unknown, it is worth speculating on the process causing the large smoothing/wall exchange observed. Especially the H2O panel in Figure S3 begs the question of where the WVISS water physically goes (probably the liquid phase).

Response: This larger smoothing effect is probably due to a large reservoir of water molecules on and in potentially in the spaces of the polymer structure and slow flow rate of water molecules through the tubing. The editor suggested that we "try to estimate what the physical size of the reservoir must be such that it can 1) exchange with all the supplied water and 2) not be isotopically changed significantly by this exchange with the gas phase water. One could calculate whether such a scenario is physically possible from the amount of gas phase water processed through the tubing.". In order to do this we estimated the reservoir size of water on/in the tubing  from a simple residence time calculation. Tau = reservoir size / exchange rate. We assumed the maximum exchange rate of water molecules with the tubing material is equal to the rate of water vapor flowing into the tubing, calculated using the flow rate and water concentration of the experiment. We used the dD location adjusted t95% as an estimate of the residence time of water on the walls, which is likely an over estimate because t66% would be closer to a tau estimate. This gave us an amount of water (in grams and weight percent) we might expect to find on the tubing walls for PTFE (0.0016 g or 0.0002%), Dekabon (0.13 g or 0.02%), and Bev-A-Line XX (1.72 g or 0.2%) to explain the long transition times. To evaluate the feasibility of that amount of water stored in the tubing, we compared this with published information. The grams of water that could possibly be absorbed by PTFE when fully submerged (water absorption % by tubing weight), <0.01%, leads to ~0.1 g H2O absorbed by 100 ft of PTFE tubing when submerged. For Dekabon and Bev-A-Line, we compared against absorption percentages for ethylene-vinyl alcohol copolymers from Cava, et al., 2006 to calculate possible grams of

water absorbed. If the reservoirs of Dekabon and Bev-A-Line XX are indeed anywhere near the maximum size calculated by Cava et al., for ethylene-vinyl alcohol copolymers (12% water absorbed by weight), and the exchange rate as slow as our tubing air flow rate, our results seem reasonable. Additionally, we wish to reiterate that we conducted this experiment at speeds below what would typically be found in atmospheric observations, so the total water molecule flow rate was slow enough to identify potential material wall effects. Lines 570-575 and 689-712 in the marked up main text were edited to reflect this discussion.

**Line 440: How or why is Dekabon dD and d18O transitions different?**

Response: Hydrogen bonding in general (referenced in lines 93-95, as well as newly expanded on in Discussion Sect. 4.5) causes differences in dD and d18O transition speeds. For Dekabon specifically, we have more clearly commented on the visual differences between Dekabon and the rest of the tubings in Figs. 4 and S4 in Sect. 3.3.1. We have also added additional calculations to the discussion of tubing type to Sect. 4.5 to address the differences in Dekabon and Bev-A-Line XX from the rest of the tubing types.

**Line 679: Try to give some indication/explanation of why such a factor of 71 (x difference between dD and d18O transition speeds) can exist (in Dekabon). What kind of process can be affecting dD so much more?**

Response: The magnitude of the hydrogen bonding effect on dD and d18O transition speeds is related to the amount of water on the tubing walls that is exchanged and the isotopic exchange rates. Slower exchange rates based on air flow conditions may be increasing the dD/d18O ratios we see in Dekabon compared to the other tubings. We have now added these ideas to Sect. 4.5. We also would like to point out that the flow rate is also a likely factor, as common rational is that increasing air flow rate through tubing decreases memory effects overall. Our experiments were conducted at a slow flow rate to magnify the memory effects and possible memory metric ratios. Differences in the humidity in such experiments may contribute to the variability of this ratio in other experiments. We have also added additional calculations to the discussion of tubing type to Sect. 4.5 to address the differences in Dekabon and Bev-A-Line XX from the rest of the tubing types.

**Small comments**

**L. 369: there is no table S2. Table S1? Optionally change the name of the supplement headers (S1-S6) to not match the figure and table names.**

The previous submission had supplemental tables in the pdf file and an excel file. We have removed Table S1 from the supplemental pdf document and have submitted it as another separate excel file, in keeping with Table S2 (originally submitted as a separate excel file). We have also renamed the excel files to start with "TableSX" in order to make this more explicit.

**L. 390: No location time in table S1/S2.**

Location time is in the Excel file Table S2.

**L. 609: amount -> margin**

Adjusted.

**L. 795: isotopic change -> the isotopic step change**

Adjusted.

**Supplemental: Figure S4 and Figure S9 seem to have accidental double axis labels.**

Adjusted.

References

Cava, D., Cabedo, L., Gimenez, E., Gavara, R., and Lagaron, J. M.: The effect of ethylene content on the interaction between ethylene–vinyl alcohol copolymers and water: (I) Application of FT-IR spectroscopy to determine transport properties and interactions in food packaging films, Polymer Testing, 25, 254–261, https://doi.org/10.1016/j.polymertesting.2005.09.018, 2006.